



# First phytoplankton community assessment of the Kong
# Håkon VII Hav, Southern Ocean during austral autumn
Hanna M. Kauko[1], Philipp Assmy[1], Ilka Peeken[2], Magdalena Różańska[3], Józef M. Wiktor[3],
Gunnar Bratbak[4], Asmita Singh[5,6], Thomas J. Ryan-Keogh[5], Sebastien Moreau[1]
[1] Norwegian Polar Institute, Fram Centre, Tromsø, Norway
[2] Alfred Wegener Institute Helmholtz Centre for Polar and Marine Research, Bremerhaven, Germany
[3] Institute of Oceanology, Polish Academy of Sciences, Sopot, Poland
[4] Department of Biological Sciences, University of Bergen, Bergen, Norway
[5] Southern Ocean Carbon and Climate Observatory (SOCCO), Council for Scientific and Industrial Research
(CSIR), Cape Town, South Africa
[6] Department of Earth Sciences, Stellenbosch University, Stellenbosch, South Africa
Correspondence: Hanna M. Kauko, hanna.kauko@npolar.no; hanna.kauko@alumni.helsinki.fi
Key words: phytoplankton, chemotaxonomy, biodiversity, Weddell Gyre, carbon and silicon cycles
Key points:
1) A typical Southern Ocean open ocean phytoplankton community dominated by heavily silicified
diatoms was observed in the Kong Håkon VII Hav in autumn 2019
2) Blooms dominated by the diatom *Chaetoceros dichaeta* were observed in two of the sampling areas
3) The other areas, mainly in a post-bloom phase, had high relative contribution from flagellates,
predominantly from the Chl *c* -lineage



**Abstract**
We studied phyto- and protozooplankton community composition based on light microscopy, flow cytometry
and photosynthetic pigment data in the Atlantic sector of the Southern Ocean during March 2019 (early austral
autumn). Sampling was focused on the area east of the prime meridian in the Kong Håkon VII Hav, including
Astrid Ridge, Maud Rise and a south-north transect at 6° E. Phytoplankton community composition throughout
the studied area was characterized by oceanic diatoms typical of the iron-deplete High-Nutrient Low-
Chlorophyll (HNLC) Southern Ocean. Topography and wind-driven iron supply likely sustained blooms
dominated by the centric diatom *Chaetoceros dichaeta* at Maud Rise and at a station north of the 6° E transect.
For the remainder of the 6° E transect diatom composition was similar to the previously mentioned bloom
stations but flagellates dominated in abundance suggesting a post-bloom situation and likely top-down control by
krill on the bloom-forming diatoms. Among flagellates, species with haptophyte-type pigments were the
dominating group. At Astrid Ridge, overall abundances were lower and pennate were more numerous than
centric diatoms, but the community composition was nevertheless typical for HNLC areas. The observations
described here show that *C. dichaeta* can form blooms beyond the background biomass level and fuels both
carbon export and upper trophic levels also within HNLC areas. This study is the first thorough assessment of
phytoplankton communities in this region and can be compared to other seasons in future studies.
**1. Introduction**
Phytoplankton play an important role for marine food webs and biogeochemical cycles as primary producers and
important mediators of the biological carbon pump. They are represented by a vast diversity of species that
occupy various ecological niches and play different ecological and biogeochemical roles, with diatoms and
haptophytes generally the main bloom-forming taxa at high latitudes (Arrigo et al., 1999; Assmy et al., 2013;
Deppeler and Davidson, 2017; Tréguer et al., 2018). Hence, for a full characterization of an ecosystem and its
biogeochemical function, it is important to investigate the phytoplankton species composition.
In the Southern Ocean, phytoplankton communities have been coarsely divided into two broad categories
(Smetacek et al., 2004). Communities characteristic of iron-replete regions such as in coastal polynyas and near
the Antarctic Peninsula and subantarctic islands (e.g. Blain et al., 2007; Pollard et al., 2009) are dominated by
bloom forming species with a 'boom and bust' life cycle and high carbon export, and largely composed of
weakly-silicified diatoms and *Phaeocystis antarctica*. The iron-limited High-Nutrient Low-Chlorophyll (HNLC)
areas of the Antarctic Circumpolar Current (ACC) on the other hand are characterized by communities
dominated by heavily silicified diatoms that largely drive the selective export of silicon (Assmy et al. 2013).
Hence the impact on biogeochemical cycles differs dramatically depending on phytoplankton community
composition. It however needs to be noted that within the diatom community representative of the iron-limited
ACC certain species can support enhanced carbon export upon relief of iron limitation (Assmy et al., 2013;
Smetacek et al., 2012). Outside of the bloom periods the community composition in areas such as the Weddell
Gyre is typically characterized by smaller cells such as haptophyte flagellates (Vernet et al., 2019). The
communities also have a varying role as prey and in the marine food webs: the large and heavily silicified
bloom-forming species can be grazed by krill but are avoided by microzooplankton grazers, which can control
the abundance of smaller prey (e.g. Irigoien et al., 2005; Löder et al., 2011; Smetacek et al., 2004).





This study was carried out as part of an ecosystem cruise in March 2019 to the Kong Håkon VII Hav, an area off
Dronning Maud Land mainly east of the prime meridian that encompasses parts of the Eastern Weddell Gyre.
The cruise observations and satellite chlorophyll *a* (Chl *a*) data have shown distinct phytoplankton phenologies
in the region, such as between Astrid Ridge and Maud Rise (Kauko et al., 2021). Knowledge on the community
composition complements our understanding of this regional variability. As Vernet et al. (2019) highlighted in
their review about the Weddell Gyre, thorough characterizations of the phytoplankton community in this area are
sparse, particularly in the area east of the prime meridian. This area is poorly studied, while spatial management
processes require improved knowledge of the ecosystem. We used different methods, with each giving a
complementary, though not complete picture of the phytoplankton community composition: light microscopy,
flow cytometry and algal pigment analysis via High Performance Liquid Chromatography (HPLC) and the
statistical method CHEMTAX (Mackey et al., 1996). The objectives of this study are to characterize the
phytoplankton and other protists communities in Kong Håkon VII Hav in late summer – early autumn, delineate
their spatial variability, and to discuss the environmental control of community composition.
**2. Methods**
**2.1 Field sampling and laboratory analyses**
The data for this study were collected during a research cruise with RV Kronprins Haakon to Kong Håkon VII
Hav, in the Atlantic sector of Southern Ocean, from February to April 2019. Sampling stations were located at
64.8 – 69.5° S and 2.3 – 13.5° E with Maud Rise, Astrid Ridge and a south-north transect at 6° E as the main
focus areas (Fig. 1). In addition, two stations were sampled in between the areas: station 53 at 68.1° S, 6.0° E and
station 54 at 68.5° S, 8.3° E. Station 53, though geographically close to the 6° E transect, showed much higher
biomass and a distinct bloom event (Kauko et al., 2021; Moreau et al., in prep.) and was therefore considered
separately.
Water samples were collected from multiple depths in the upper 100 m at a total of 37 stations (station numbers
starting with 53) between 12 and 31 March in connection with CTD (conductivity-temperature-depth) casts with
a 24-bottle or 12-bottle SBE 32 carousel water sampler.
Samples for phytoplankton microscopy analyses (190 mL) were collected from 3 different depths (typically 10,
25 or 40, and 75 m), filled into 200 mL brown glass bottles and fixed with glutaraldehyde and 20%
hexamethylenetetramine-buffered formaldehyde at final concentrations of 0.1 and 1%, respectively, and
thereafter stored cool and dark. For analysis, 10–50 mL subsample were settled in Utermöhl sedimentation
chambers (HYDRO-BIOS©, Kiel, Germany) for 48 h and counted with a Nikon Ti-U inverted light microscope
using the Utermöhl method (Edler and Elbrächter, 2010). Protists cells were counted in fields of view located
along transects crossing the bottom of the chamber. In each sample, at least 50 cells of the dominant species
were counted (error of ±28% according to Edler and Elbrächter, 2010).
Flow cytometry (FCM) samples (4.5 mL) for counting cells in small algal size classes (pico- and
nanophytoplankton, 0.7 to 2 µm and 2 to 20 µm, respectively) were collected in cryovials from 5-6 different
depths, fixed with glutaraldehyde (0.5% final concentration) and stored in -80° C until analyses at the University
of Bergen. In the laboratory, samples were thawed, mixed gently, and analysed in an Attune™ NxT Acoustic





Focusing Cytometer (Invitrogen™, Thermo Fisher Scientific Inc. USA) equipped with a 50 mW 488 nm (blue)
laser. Quantification and discrimination of the different phytoplankton size classes was done with the help of
biparametric plots based on side scatter and red fluorescence.
Samples for algal pigment analysis (usually 1 L) were collected from 3 different depths (typically 10, 25 or 40,
and 75 m), filtered on 0.7 µm GF/F filters (GE Healthcare, Little Chalfont, UK) with a gentle vacuum pressure
(approximately −30 kPa), and immediately stored in the dark at -80° C. Pigments were measured and quantified
with a Waters Alliance 2695 HPLC Separation Module connected to a Waters photodiode array detector (2,996).
HPLC-grade solvents (Merck) and an Agilent Technologies Microsorb-MV3 C8 column (4.6 × 100 mm) was
used for peak separation. The auto sampler module was kept at 4°C during the measurements. In total 100 µl
sample were injected with an auto addition function of the system between sample and a 1 molar ammonium
acetate solution in the ratio of 30:20:30:20. Peak identification and quantification was obtained with the
EMPOWER software. More details about the solvents and gradient can be found in Tran et al. (2013). Overview
of the taxonomical distribution of pigments is given in Jeffrey et al. (2011), Higgins et al. (2011) and the data
sheets of Roy et al. (2011).





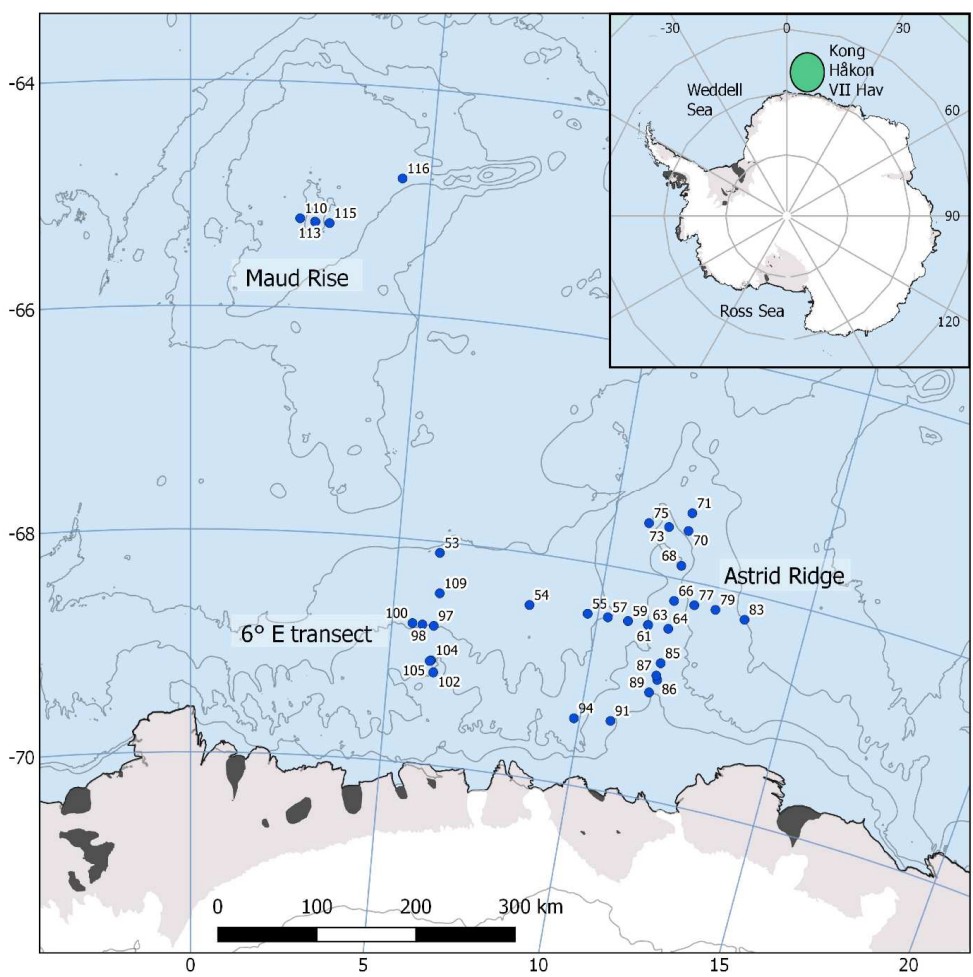


**Figure 1: Map of the study area. The CTD stations with water sampling are marked with blue circles. The sampling area is marked with a green ellipse in the insert. Map created with the help of Quantarctica (Norwegian Polar Institute, 2018).**


**2.2 Statistical analyses**
Similarity between the sampling areas in terms of the microscopy counts was evaluated with non-metric
multidimensional scaling (NMDS) using the *isoMDS* function in the MASS package (Venables and Ripley,
2002) and the R software (R Core Team, 2017). CTD samples down to 100 m depth with full taxonomical
resolution were used for the analysis. Bray-Curtis dissimilarities (vegan package in R; Oksanen et al., 2017)
were used for the scaling and abundances were square-root transformed prior to that to reduce the effect of high
and uneven abundances. The dissimilarities between the groups were further tested statistically with the *anosim*
function from the vegan package. Test result values (R values) close to 0, as opposed to 1, indicate random
grouping. For the test considering differences between the sampling areas, the assumptions of heterogeneity and



similar sample size were not met, however, due to the lower range of dissimilarities occurring in the smaller-
sized sample group Maud Rise (Fig. A1), the test tends to be overly conservative (Anderson and Walsh, 2013)
and thus a significant result appears reliable.
Diversity in the phytoplankton community was investigated with the Shannon's diversity index (H; function
*diversity* in the vegan package) and species richness (number of species, genera and size groups of unidentified
taxa). Differences between the areas and sampling depths were tested with one-way Analysis of Variance
(ANOVA; function *aov* in R). The assumptions of homoscedasticity were met in the models.

### 2.3 CHEMTAX analysis

Phytoplankton community composition was further investigated by applying a factor analysis program called
CHEMTAX (Mackey et al., 1996), which allows to calculate the abundance of the various algal groups based on
the measured marker pigments. As we had a large number of samples and no experimental or field information
on local pigment ratios, the original approach (Mackey et al., 1996) was concluded to be more suitable than the
Bayesian approach (Van den Meersche et al., 2008), according to Higgins et al. (2011). The software package
CHEMTAX was obtained from Wright (2008).
The initial ratio matrix was based on literature. Pigment to Chl *a* ratios for prasinophytes, chlorophytes,
cryptophytes, two pigment types of diatoms and peridinin-containing dinoflagellates were taken from the table in
Wright et al. (2010), a study that was conducted close to our study area (between 30° to 80° E and south of 62°
S), with the following modifications. Chl $c_1$ was changed to Chl $c_{1+2}$ (which is the resolution of our
chromatographic results) with values taken from the CHEMTAX material (geometric means of reported ratios
from the literature collected in Higgins et al., 2011). The values for 19'-butanoyloxyfucoxanthin (but-fuco),
ratios for haptophytes pigment type 6 and for dinoflagellates pigment type 2 (microscopy revealed dominance of
*Gymnodinium* spp.) were taken from Table 6.1 in Higgins et al. (2011). Zeaxanthin was observed in only one
sample and was omitted from the analysis. Diadinoxanthin, diatoxanthin and β,β-carotene were excluded
because they are not very group-specific. Neoxanthin, prasinoxanthin and violaxanthin were not observed in the
samples and were removed from the ratio matrix.
Haptophytes belong to several (8) different pigment types (Zapata et al., 2004) and in addition change their
marker pigment content according to environmental conditions such as iron availability (van Leeuwe and Stefels,
1998; Wright et al., 2010). Therefore, all haptophyte pigment types were initially tested with CHEMTAX runs
on all samples (20 randomized ratio matrices, using the pigment ratios from the CHEMTAX material mentioned
above as initial ratios). The pigment type 8 is typical in the Southern Ocean including the species *P. antarctica*,
whereas coccolithophores belong to pigment type 6. Out of the eight different pigment types tested, including
pigment types 6, 7 or 8 resulted in the lowest root mean square errors (RMSE; below 0.2). Pigment type 7
includes e.g. the genus *Chrysochromulina* which is not typical in the Southern Ocean. Including both haptophyte
type 6 and 8 (in different ratio range categories according to the CHEMTAX instructions) also resulted in a low
RMSE, and for the categories with high ratio range for haptophyte type 6 the error was lowest and similar to
when including only haptophyte type 6 (<0.15). However, coccolithophores should not be abundant this far
south (Balch et al., 2016; Saavedra-Pellitero et al., 2014; Trull et al., 2018) and were not observed in the
microscopy samples. Other prymnesiophytes were not abundant either – only *P. antarctica* was observed in only



three CTD samples. This taxon has a characteristic appearance and, if present in large quantities, would likely
have been identified, whereas the majority of flagellates in the microscopy samples were classified as
unidentified flagellates in the 3 to 7 µm size range. Therefore, to simplify the analysis (e.g. to avoid having too
many algal groups compared to pigments, Mackey et al., 1996) and to account for the unidentified status of this
group, we have included only one haptophyte group in the final runs with the best-performing i.e. type 6 pigment
ratios and called this "Haptophytes-6 -like". Silicoflagellates and chrysophytes, that were observed at low
abundances in microscopy samples (maximum abundances of 3900 and 18200 cells L$^{-1}$, respectively), will also
be included in the haptophyte pigment group, as they contain similar pigments, e.g., Chl *c*, fucoxanthin and its
derivatives (Jeffrey et al., 2011).
In the preliminary analysis, it was also tested to separate the samples into different clusters. With all samples
combined, including only the surface samples down to 10 m, or successively adding depth ranges one at a time
did not improve the result in terms of the RMSE, compared to including all depths. Separating Maud Rise from
the rest reduced the error, when different area clusters were tested with all samples. Trials indicated that dividing
the Maud Rise samples into depth clusters may bring further improvements but as the number of samples was
relatively small (in total 12 CTD samples from Maud Rise) they were kept as one cluster. Astrid Ridge had a
larger number of samples (55 in total) and was divided into two clusters (above and below including 40 m;
average mixed layer depth (MLD) was 34 m, Kauko et al., 2021) and separated from the rest, which reduced the
error. For the 6° E transect, separating the surface samples did not reduce the error.
In total there were 98 samples from the CTD casts. In the clusters Maud Rise, Astrid Ridge surface, Astrid Ridge
deep and other stations (stations 53, 54 and 6° E transect) there were 12, 26, 29 and 31 samples, respectively.
After the 60 first runs for each of the clusters (using 60 randomized pigment ratio matrices based on the initial
ratio matrix), the average output ratio matrix of the 6 best runs was used as the initial ratio matrix for the next 60
runs. The reported results are the averaged output from the six best runs of this second step.
**3. Results**
**3.1 Microscopy**
The microscopy data are shown here as averages per sampling area and for the most important taxa separately,
whereas others are summed together into higher-level categories such as "Pennate diatoms (other)". All taxa are
listed in Table B1 together with median abundances and occurrence in the different sampling areas, and variance
in data used for the averages (i.e., data from all samples) is shown in Fig. A2 and A3.
Two of the sampling locations had an active diatom bloom, with average diatom abundances at station 53 and
Maud Rise reaching $5.2 \times 10^5$ and $7.5 \times 10^5$ cells L$^{-1}$, respectively (Fig. 2a), and Chl *a* data showing the highest
biomass in the area (Fig. 3; Kauko et al., 2021). Most of the sampling areas were dominated by diatoms in terms
of average abundances, most notably for the area represented by station 53 and Maud Rise (74 and 89 %,
respectively), whereas at station 54 or Astrid Ridge the dominance was less pronounced (62 and 56 %), and the
area along the 6° E transect was slightly dominated by flagellates (45 % flagellates compared to 36 % diatoms).
At Maud Rise flagellates and dinoflagellates occurred in similar abundances whereas in the other areas,
flagellates were more abundant than dinoflagellates, most notably so along the 6° E transect. Ciliates and





cyanobacteria (unidentified filamentous blue-green algae cf. *Anabaena* sp., see photo in Fig. A4) were also

observed at very low abundances, especially the latter mainly at Astrid Ridge and along the 6° E transect. FCM

biplots (Fig. A5) using orange fluorescence indicated the presence of cyanobacteria in the corresponding

samples, however abundances were low and the filamentous nature of the cyanobacteria complicates

interpretations for this method.

The dominance patterns were similar when abundances were averaged per depth interval (Fig. A6), but at Astrid

Ridge diatoms formed less than half of the community (about 30 %) below 45 m where dinoflagellates were

slightly more prominent (32 to 37 %). In contrast, along the 6° E transect diatoms dominated at 75 m and formed

about half of the community at 50 m. In terms of abundances, phytoplankton were concentrated in the upper 40

m at station 53 and Astrid Ridge, whereas along the 6° E transect the generally low abundances were more

evenly distributed with depth and at Maud Rise the bloom extended deeper with relatively high cell numbers (4

$\times 10^5$ cells L$^{-1}$) until 75 m.

Among the diatoms, *Chaetoceros dichaeta* clearly dominated station 53 and Maud Rise communities down to 40

and 50 m, respectively (Fig. 2b-c, 4 and A7). *Chaetoceros dichaeta* formed 59 % of the diatom community at 10

m and 40 % at 40 m at station 53, i.e. it was the most abundant species at these depths. At Maud Rise, besides

the surface samples, *C. dichaeta* dominated the diatom community at 100 m depth (at station 110; Fig. A8). This

species was also an important component of the 6° E transect diatom community although at much lower

abundances. In these other sampling areas not characterized by an active bloom (the 6° E transect, station 54 and

Astrid Ridge), the abundances of various diatom species were more evenly distributed. Other important taxa

were *Fragilariopsis* spp., *F. nana, F. kerguelensis, F. cylindrus, Dactyliosolen antarcticus, Chaetoceros* spp.

and *Pseudo-nitzschia* spp. At Astrid Ridge and station 54, pennate diatoms (particularly *Fragilariopsis* spp. and

*Pseudo-nitzschia* spp.) were more abundant than centric diatoms, with shares of 72 and 56 %, respectively. In

other areas pennate diatoms contributed 14 to 34 %. Overall, there were 89 diatom taxa (at the genus or species

level) identified during this research campaign.

Maximum average abundances of flagellates were observed at station 53 and along the 6° E transect, with $1.0 \times$

$10^5$ and $1.1 \times 10^5$ cells L$^{-1}$, respectively (Fig. 2d). Among the flagellates, a majority was categorized as

unidentified flagellates in the size range 3 to 7 µm. Cryptophytes and especially the genus *Telonema* were also a

notable component of the flagellate community in many of the areas (in Fig. 2d cryptophytes and the genus

*Telonema* are presented separately). Choanoflagellates (heterotrophic flagellates) were observed at relatively

high numbers at station 53 and Maud Rise. *Phaeocystis antarctica* (the only prymnesiophyte species identified)

was found at station 54 mainly at 40 m, but it was not an abundant species during the cruise, which was also

confirmed by microscope analysis of live material from net samples taken from the upper 20 m at every CTD

station during the cruise. Chlorophytes, chrysophytes, prasinophytes and silicoflagellates were also observed in

minor numbers. The depth distribution of flagellates (figures not shown) was largely similar to the composition

of the whole area averages, but choanoflagellates were most prominent at 25 m at Maud Rise.

Dinoflagellates belonged mainly to different, unidentified species of the genus *Gymnodinium* in all areas (Fig.

2e) and at all depths (figures not shown). Additionally, the genera *Prorocentrum, Gyrodinium, Alexandrium,*

*Amphidinium, Polarella* and *Protoperidinium* were also present. The maximum average dinoflagellates

abundance was observed at station 53 ($8.2 \times 10^4$ cells L$^{-1}$).



Ciliates were present in lower numbers (the maximum average abundance was 1500 cells L$^{-1}$ at Maud Rise; Fig.
2f) but with several species (16 species or higher level taxa; Table B1). The most notable species were
*Salpingella costata, Strombidium* spp., and *Lohmanniella oviformis*, as well as *Uronema marinum* at station 53
and *Mesodinium rubrum* at station 54. At Astrid Ridge and along the 6° E transect, aloricate (naked) ciliates
dominated in abundance (at station 54 the dominance was less pronounced), whereas at Maud Rise the
abundances were even and at station 53 loricate ciliates (tintinnids) dominated (Fig. 2f). Ciliate abundances were
lowest at station 54 (125 cells L$^{-1}$).





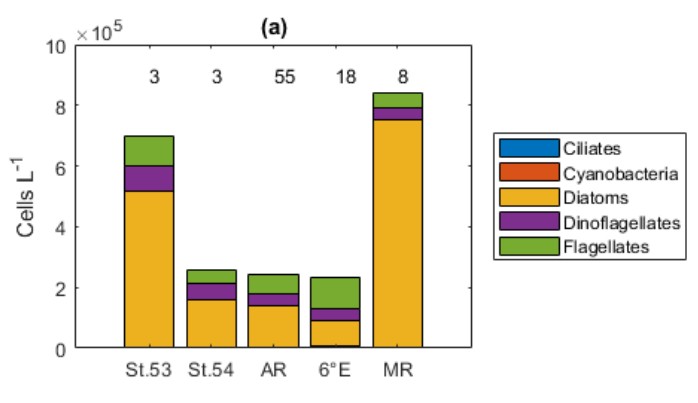

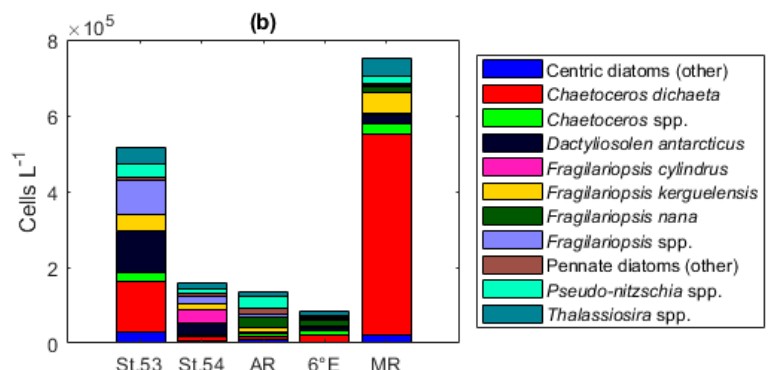

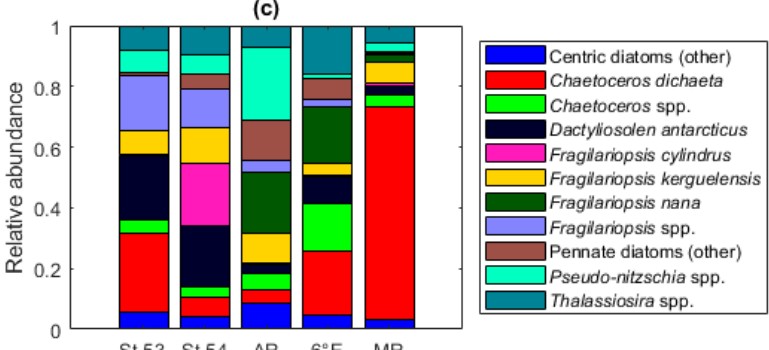


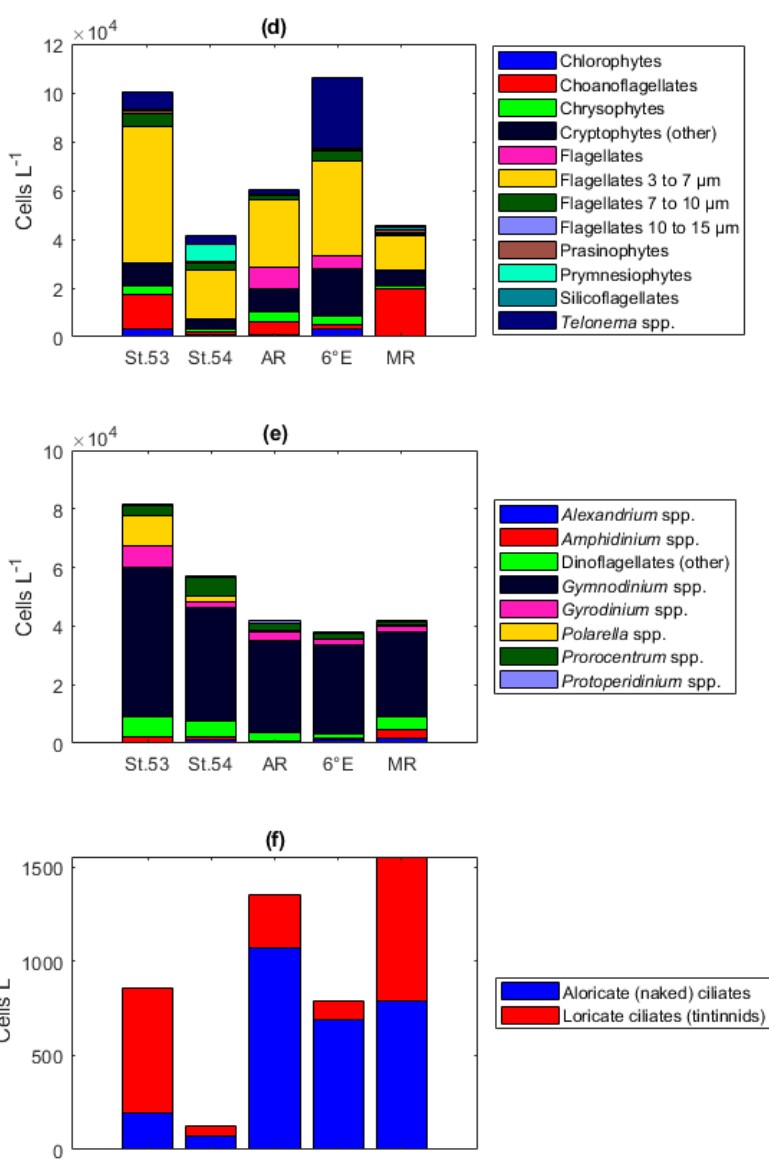

**Figure 2: Abundance of different protist groups and species for (a) main taxa, (b) diatoms, (c) relative abundance of diatoms, (d) flagellates, (e) dinoflagellates and (f) ciliates. In (a), the number of samples used for the average abundances is shown in the top of the figure (the numbers apply to all figures). In (c) and (d), the genera *Fragilariopsis* and *Pseudo-nitzschia* belong to pennate diatoms, thus pennate diatoms are shown with colours pink/yellow to cyan. St.53=station 53, St.54=station 54, AR=Astrid Ridge, 6E=6° E transect, MR=Maud Rise.**



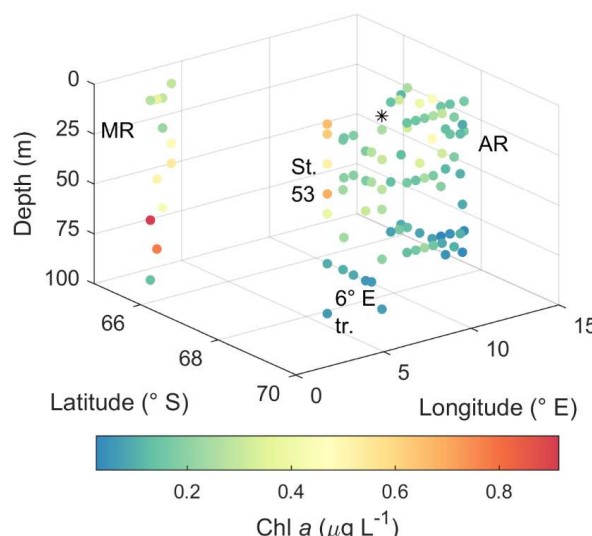


**Figure 3: Horizontal and vertical distribution of phytoplankton biomass expressed as Chl *a* concentration. MR=Maud Rise, St. 53=station 53, AR=Astrid Ridge, 6° E tr.= 6° E transect. Station 54 is marked with a black asterisk.**



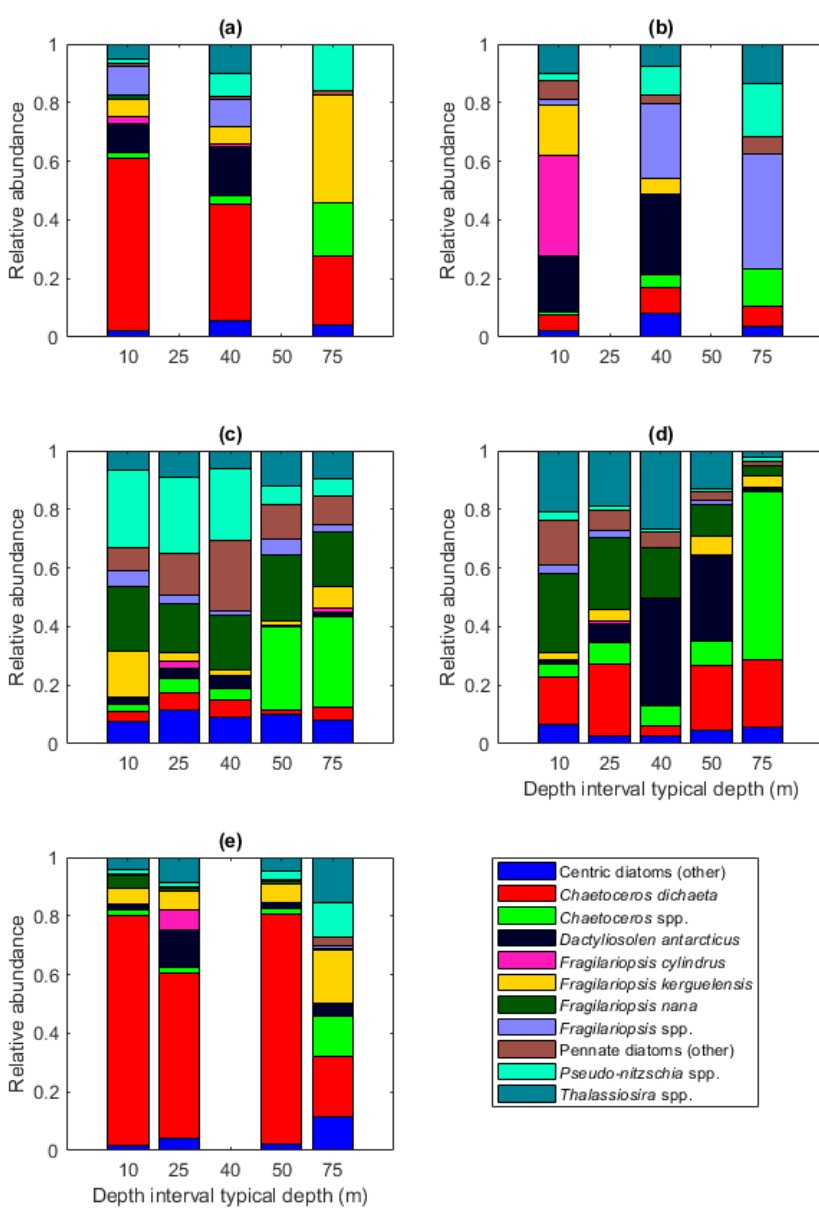

**Figure 4: Diatom relative abundance in the different sampling areas averaged per depth interval for (a) station 53, (b) station 54, (c) Astrid Ridge, (d) 6° E transect and (e) Maud Rise. Depth intervals (with typical sampling depth in brackets): 5-10 (10); 25-35 (25), 35-45 (40), 50-60, 65-85 (75) m.**







Clustering (NMDS) of the abundance results from the microscopy analysis showed that the communities in the
different sampling areas (marked with different symbols in Fig. 5) did not separate into distinct clusters, but they
appear located at different sides of the cluster, with station 53 and 54 and Maud Rise samples on one side and the
Astrid Ridge and 6° E transect samples predominantly on the other side. In addition to the diatom blooms in the
first two mentioned areas, this could also reflect a coastal to offshore pattern. However, the low R value of 0.15
from the *anosim* test (significance 0.017) indicated overall a high similarity between the areas.
In addition, a separation along the sampling depth gradient (colour scale in Fig. 5) is clearly visible, with the
surface samples (typically sampled at 25 m depth) and the deep samples (typically sampled at 75 m depth)
located on different sides of the cluster. The *anosim* test indicated a somewhat higher degree of differentiation
between the depth clusters (R value 0.27, significance 0.001) than between the sampling areas. In addition, when
the NMDS analysis is performed on presence-absence data (Fig. A9), it is difficult to separate the areas, but the
sampling depth pattern is still visible, though the samples are very condensed on the plot. Other categorizations
included in the analysis, such as according to bottom depth, latitude or separation of Astrid Ridge into different
areas (north, south, west and east parts of the Ridge), did not yield such clear patterns (figures not shown).
The Shannon's diversity index varied between 0.9 and 3.4, and the species richness between 11 and 65
species/taxa. The biodiversity between the areas was relatively similar, but the most notable geographical
patterns were that most depths at Maud Rise had a low diversity index, and that species richness in the other
sampling areas was lower at depth than in the upper part of the water column (Fig. 6a and b). This was also
visible in the statistical analysis of differences between groups: regarding the diversity index, the differences
between areas were highly significant (p-value <0.001), but not between depth categories (p-value 0.32; the
same depth categories were used as in the Fig. 4). A post-hoc Tukey test confirmed that Maud Rise differed from
all other areas (p-value <0.02 for all comparisons). For species richness the inverse was found, differences
between depth categories were significant (p-value <0.001) and not between the areas (0.69). A post-hoc Tukey
test showed that the surface depth categories (10, 25 and 40 m) differed from the deeper categories (50 and 75 m;
p-value for all comparisons <0.02, except for between 50 and 25 m where the p-value was 0.06), that is, species
richness was significantly lower at depth (50 m and deeper). The means for the different areas were 2.7, 3.0, 2.7,
2.6 and 1.9 for the diversity index and 49, 47, 44, 45 and 49 for species richness for station 53, station 54, Astrid
Ridge, 6° E transect and Maud Rise, respectively. The mean diversity index was thus significantly lower at
Maud Rise. The diversity index did not have a clear correlation with biomass, but species richness increased with
increasing biomass up to maximum values of around 55–65 (Fig. 6c and d).

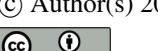



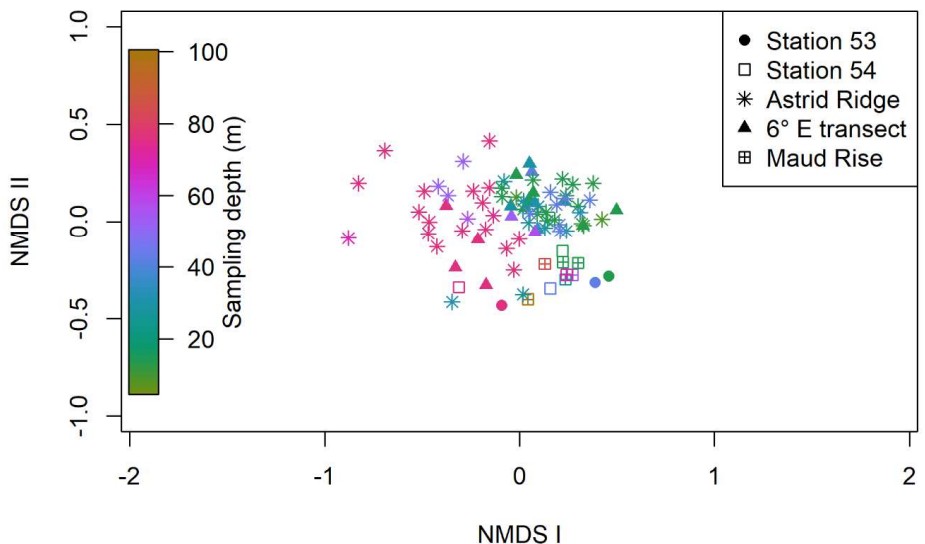


**Figure 5: Results of the NMDS clustering of the microscopy count samples. The colour shows the sampling depth and**
**the different sampling areas are shown with different symbols, see legend. The stress value of the plot is 22 %.**



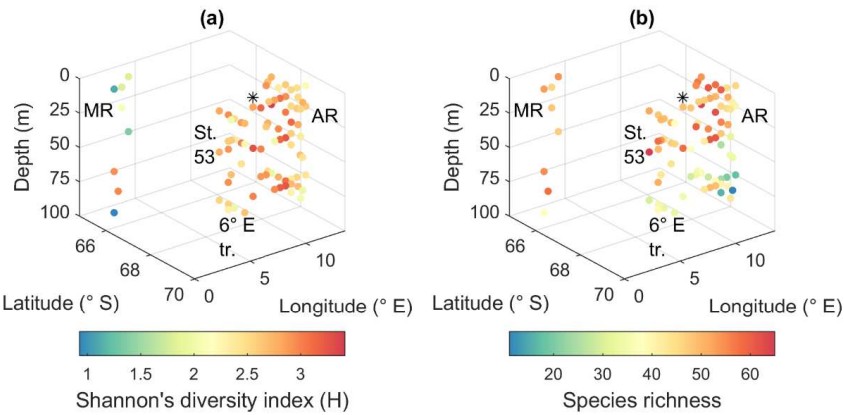

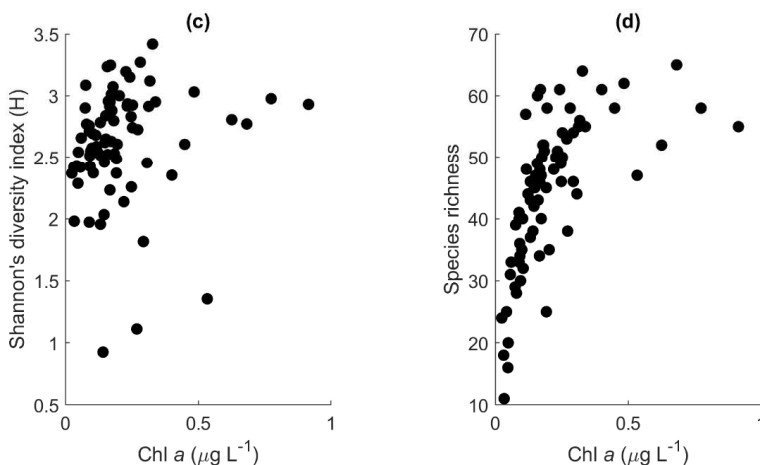


**Figure 6: Biodiversity according to the microscopy samples. (a) Shannon's diversity index, (b) species richness, (c) relationship between algal biomass (expressed in Chl a concentration) and Shannon's diversity index and (d) algal biomass and species richness. MR=Maud Rise, St. 53=station 53, AR=Astrid Ridge, 6° E tr.= 6° E transect. Station 54 is marked with a black asterisk.**

**3.2 Flow cytometry**

Smaller nanophytoplankton (Nanophytoplankton 1; Fig. A5) showed the highest abundances along the 6° E

transect, with abundances up to $4.7 \times 10^6$ cells L$^{-1}$ (Fig. 7a), and lowest at Maud Rise. On the contrary, larger

nanophytoplankton (Nanophytoplankton 2) were associated with Maud Rise and station 53 (up to $4.2 \times 10^6$ cells



$L^{-1}$; Fig. 7b). Maud Rise had high abundances also at depth, contrary to station 53. Some larger cells
(Nanophytoplankton 2) were also observed on top of Astrid ridge (stations 66, 68 and 73), near the surface.
Picophytoplankton abundance was lower than for nanophytoplankton (up to $0.7 \times 10^6$ cells $L^{-1}$; Fig. 7c), but a
few stations on the west side of Astrid ridge (57, 59, 61) showed a distinct picophytoplankton population in the
FCM biplots (Fig. A5).
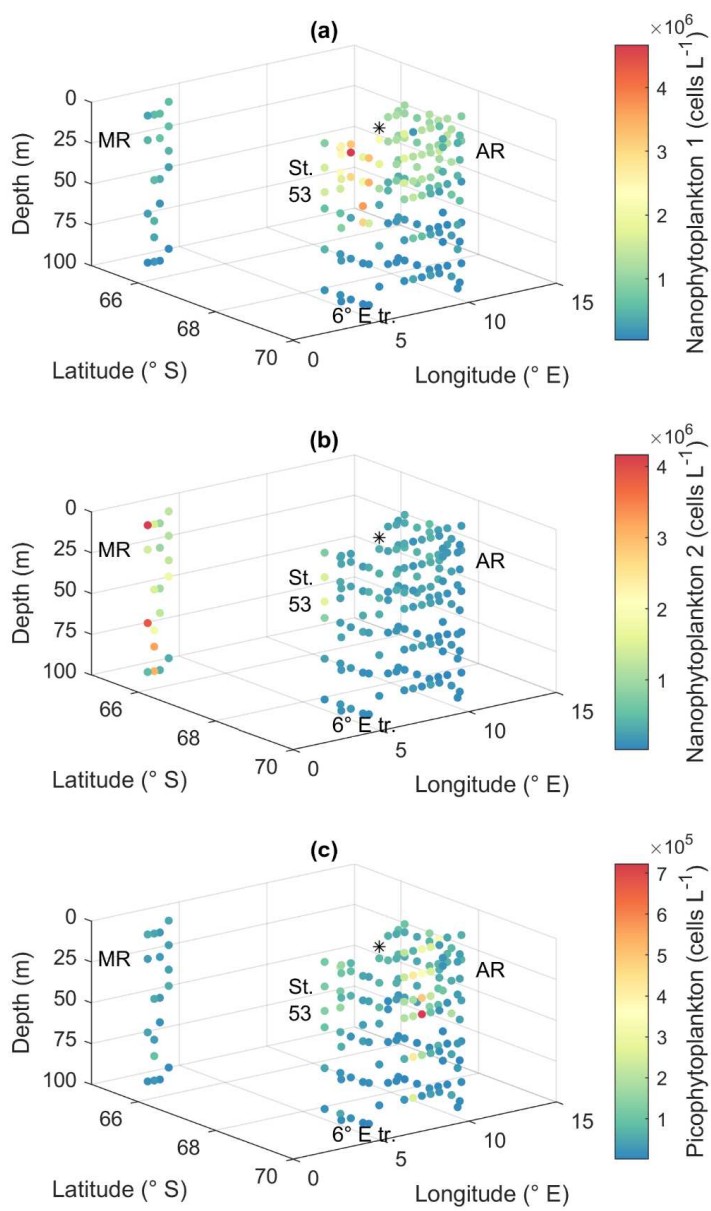


**Figure 7: Flow cytometry results. Cell abundances of two groups of nanophytoplankton (a, b) and picophytoplankton**
**(c). MR=Maud Rise, St. 53=station 53, AR=Astrid Ridge, 6° E tr.= 6° E transect. Station 54 is marked with a black**
**asterisk.**



### 3.3 Marker pigments

Pigment to Chl *a* ratios are presented in Fig. 8 and 9 and reported here, whereas the pigment concentrations are shown in Fig. A10 and A11. Chl *a* concentration ranged between 0.02 and 0.92 µg L$^{-1}$ (Fig. 3). The diatom blooms at Maud Rise and station 53, and the importance of flagellates at the 6° E transect were also visible in the pigment data.

Ratios of fucoxanthin, a typical pigment in diatoms, to Chl *a* were very high at Maud Rise and station 53, up to 0.93 (Fig. 8a). The ratios were the lowest at the 6° E transect, with a minimum of 0.12. At Astrid Ridge the ratios were in between these values at around 0.5. The ratios of Chl $c_{1+2}$ to Chl *a* were also the highest at Maud Rise and station 53, up to 0.70 and seemed thus to be primarily associated with fucoxanthin and diatoms (Fig. 8b). However, other Chl $c_{1+2}$ containing groups were also likely present, as the ratios at the flagellate-dominated 6° E transect did not differ from the other areas as much as for fucoxanthin.

Chl $c_3$ showed the highest pigment to Chl *a* ratio values at the 6° E transect and at depth at Astrid Ridge, up to 0.55 (Fig. 8c). It was also found at Maud Rise at all depths, in the surface waters at station 53 and station 54, and at Astrid Ridge mainly in the middle of the ridge, from the surface to mid-depths. This pigment thus further indicates that flagellates were an important part of the 6° E transect community, as it is a major pigment e.g. in haptophytes. In addition, 19'-hexanoyloxyfucoxanthin (hex-fuco), another important pigment in haptophytes, showed clearly its highest pigment to Chl *a* ratio values at the 6° E transect, up to 1.01, and the lowest at Maud Rise (Fig. 8d). Another fucoxanthin derivative, but-fuco, that is mainly found in pelagophytes, silicoflagellates and some haptophytes, showed the highest pigment to Chl *a* ratio values at depth at the 6° E transect and Astrid Ridge, but values were low (Fig. 8e).

Diadinoxanthin, a carotenoid participating in the photoprotective xanthophyll cycle, occurred in the highest pigment to Chl *a* ratios close to the surface in all areas (up to 0.25), but at Maud Rise relatively high ratios were observed throughout the sampling depths (Fig. 8f). Diatoxanthin, its counterpart in the xanthophyll cycle, was observed in five samples at a much lower concentration (5–16 % of diadinoxanthin). It should be noted that although the samples were processed as quickly as possible, they were part of a larger sampling effort, and conversion from diatoxanthin to diadinoxanthin may have happened during the storage under dark conditions.

Peridinin (a major pigment in one of the dinoflagellate pigment classes), alloxanthin (a major pigment in cryptophytes), lutein (Chl *b*-lineage, e.g. chlorophytes and prasinophytes) and Chl *b* were observed in minor amounts in certain areas (Fig. 9): peridinin on the west side of Astrid Ridge (pigment to Chl *a* ratio up to 0.15), alloxanthin at the surface at a few stations of the 6° E transect and Astrid Ridge (up to 0.01), and lutein and Chl *b* at the 6° E transect (up to 0.04 and 0.06, respectively). β,β-carotene is not very taxon-specific and did not show clear geographical patterns (pigment to Chl *a* ratio up to 0.05; Fig. A12). Zeaxanthin was only observed in one sample, in the surface (5 m) at station 70 at Astrid Ridge, in low concentration (ratio to Chl *a* was 0.02).





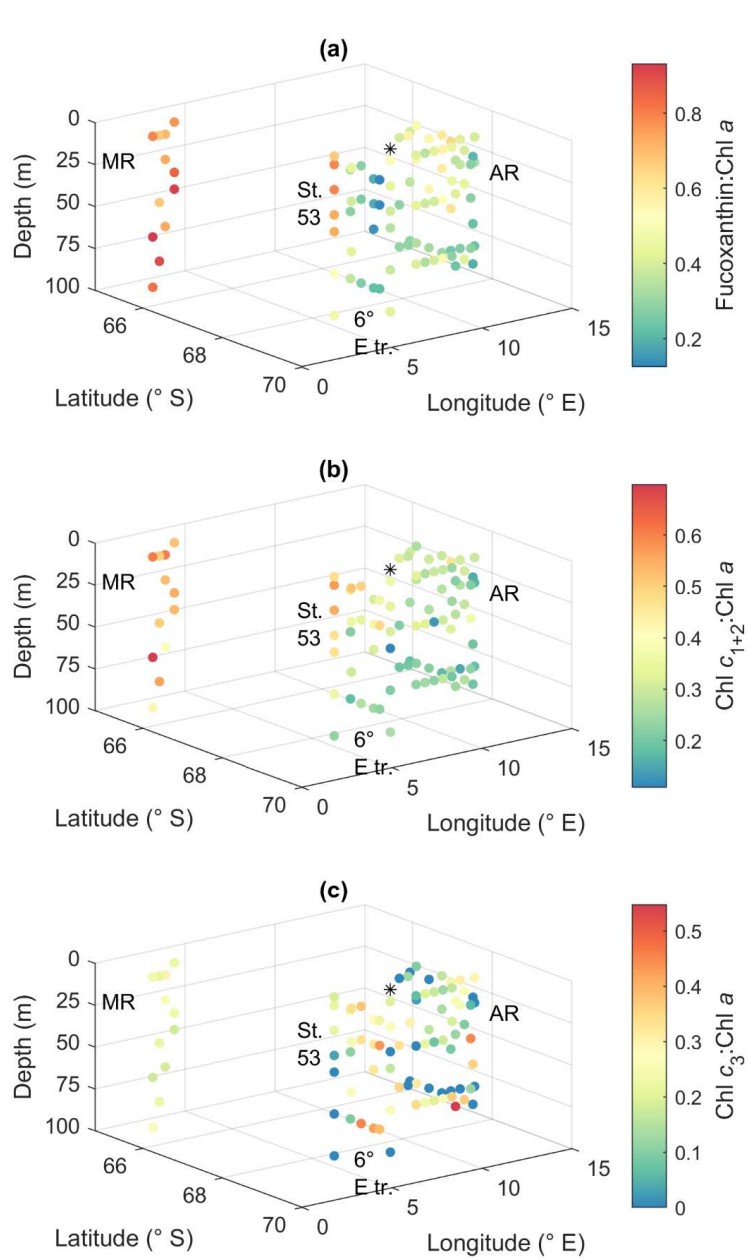






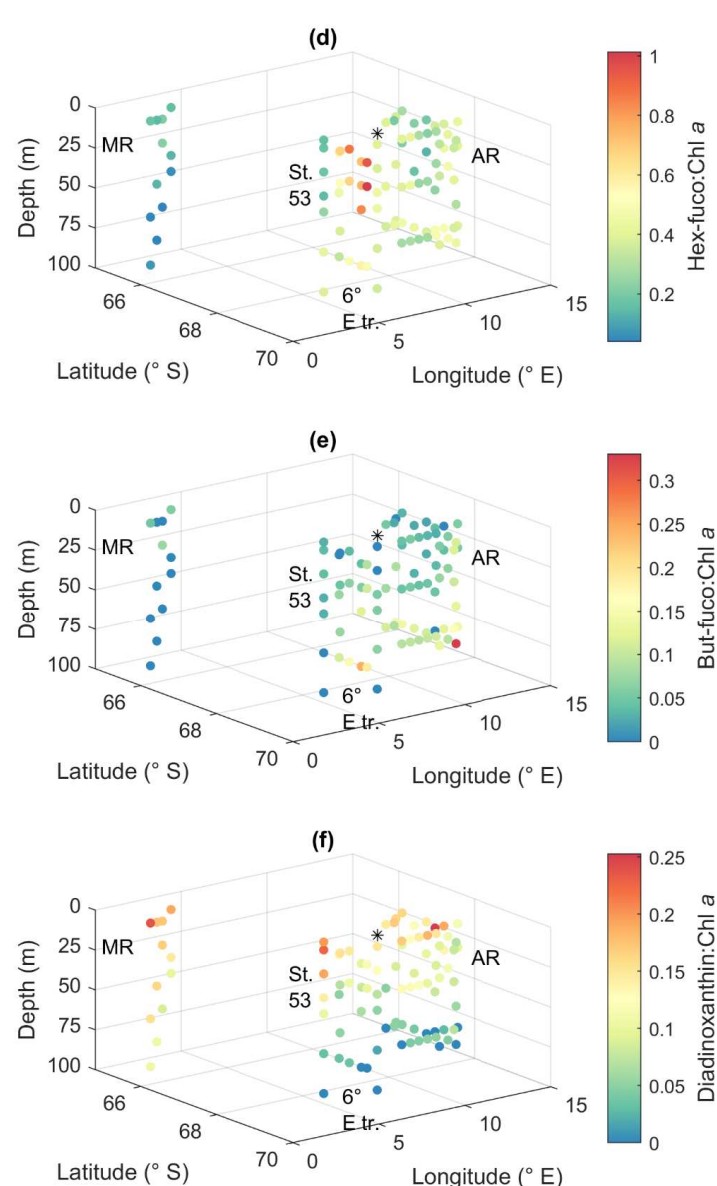


**Figure 8: Ratios of algal pigments to Chl a for (a) fucoxanthin, (b) Chl c₁₊₂, (c) Chl c₃, (d) hex-fuco, (e) but-fuco and (f) diadinoxanthin. MR=Maud Rise, St. 53=station 53, AR=Astrid Ridge, 6° E tr.= 6° E transect. Station 54 is marked with a black asterisk.**

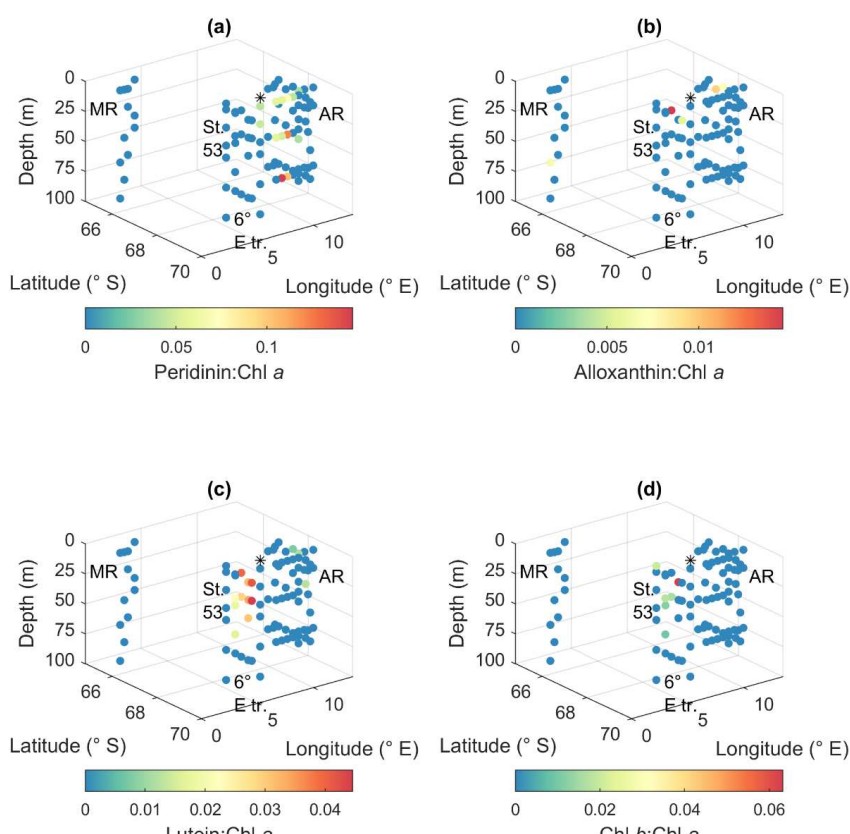


**Figure 9: Ratios of algal pigments to Chl a for (a) peridinin, (b) alloxanthin, (c) lutein and (d) Chl b. MR=Maud Rise,**
**St. 53=station 53, AR=Astrid Ridge, 6° E tr.= 6° E transect. Station 54 is marked with a black asterisk.**

## 3.4 CHEMTAX analysis

The CHEMTAX analysis is a way to distinguish and quantify the contribution of various phytoplankton groups
based on the measured marker pigment concentrations. In total eight phytoplankton groups were included in the
analysis based on prior knowledge from the microscopy results and the literature. Clear geographical patterns
were observed in the distribution of the groups in line with the other phytoplankton data sources. Diatoms
pigment type 2 (diatoms containing Chl $c_3$) had the highest biomass, followed by diatoms type 1 and the
haptophyte-like group (Fig. 10). Diatoms type 1 ranged up to 0.17 µg Chl $a$ L$^{-1}$ and had the highest values in the
upper water column at Astrid Ridge and Maud Rise. Diatoms type 2 were most prominent at station 53 and at
depth at Maud Rise with a maximum value of 0.78 µg Chl $a$ L$^{-1}$. The haptophytes-6 -like had the highest values
at Maud Rise and the upper water column at the 6° E transect with a maximum value of 0.18 µg Chl $a$ L$^{-1}$, but



clear presence also at Astrid Ridge. Of the dinoflagellate groups, type 2 had higher biomass and was present in
all areas, though only at the surface at Maud Rise, with a maximum value of 0.10 µg Chl $a$ L$^{-1}$. Occurrence of
dinoflagellates type 1 (peridinin-containing dinoflagellates), prasinophytes, chlorophytes and cryptophytes in the
CHEMTAX results (Fig. 10) followed closely the distribution of their respective marker pigments (Fig. 9) and
was correspondingly scattered and scarce. A maximum value of 0.04 µg Chl $a$ L$^{-1}$ was found for dinoflagellates
type 1 and 0.03 µg Chl $a$ L$^{-1}$ for the other three groups. From the Chl $b$-containing groups, chlorophytes were
more abundant than prasinophytes with a clear presence along the 6° E transect.
The final RMSE for the clusters Maud Rise, Astrid Ridge surface, Astrid Ridge deep and other stations (stations
53, 54 and 6° transect) was 0.017, 0.064, 0.080 and 0.069, respectively (average RMSE of the best 6 runs). The
final output ratio matrices for each of the clusters are presented in Table 1 for potential use as initial ratio
matrices in future studies in the area. It is noteworthy that differentiating the data between the sampling areas,
and in some cases along the depth gradient, improved the results.

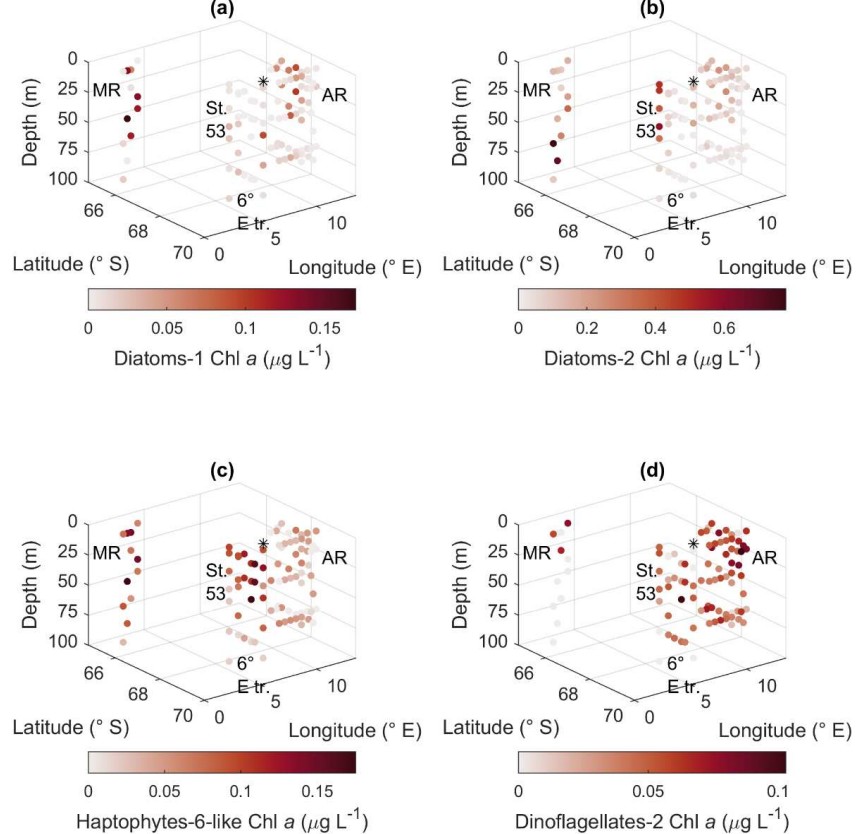






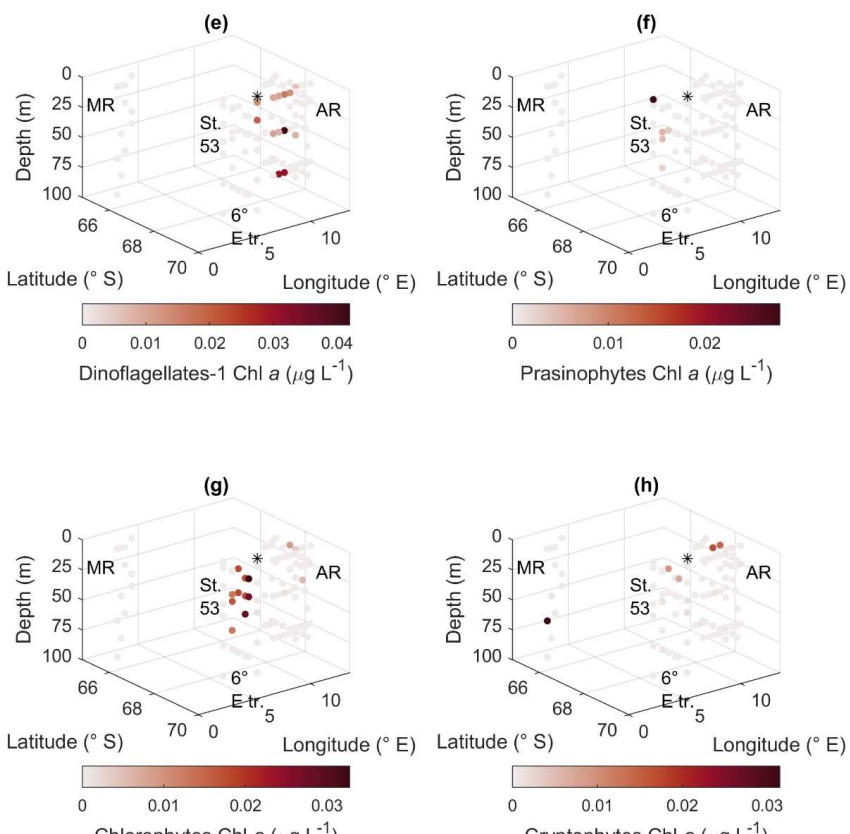


**Figure 10: CHEMTAX results for the different algal groups. (a) Diatoms type 1, (b) diatoms type 2, (c) haptophytes type 6 -like, (d) dinoflagellates type 1, (e) dinoflagellates type 2, (f) prasinophytes, (g) chlorophytes and (h) cryptophytes. MR=Maud Rise, St. 53=station 53, AR=Astrid Ridge, 6° E tr.= 6° E transect. Station 54 is marked with a black asterisk.**





**Table 1 Initial pigment to Chl a ratios used in the CHEMTAX analysis and the final ratio matrices for each cluster**
**(average of the 6 best performing runs of the second step; see Methods).**

| Initial ratios | Chl_c3 | Chlc_1-2 | Peri | But-fuco | Fuco | Hex-fuco | Allo | Lut | Chl_b | Chl_a |
|---|---|---|---|---|---|---|---|---|---|---|
| Prasinophytes | 0 | 0 | 0 | 0 | 0 | 0 | 0 | 0.0066 | 0.55 | 1 |
| Chlorophytes | 0 | 0 | 0 | 0 | 0 | 0 | 0 | 0.23 | 0.15 | 1 |
| Cryptophytes | 0 | 0.17 | 0 | 0 | 0 | 0 | 0.21 | 0 | 0 | 1 |
| Diatoms-1 | 0 | 0.09 | 0 | 0 | 1.04 | 0 | 0 | 0 | 0 | 1 |
| Diatoms-2 | 0.016 | 0.22 | 0 | 0 | 0.83 | 0 | 0 | 0 | 0 | 1 |
| Dinoflagellates-1 | 0 | 0.23 | 0.82 | 0 | 0 | 0 | 0 | 0 | 0 | 1 |
| Dinoflagellates-2 | 0.04 | 0.12 | 0 | 0.06 | 0.19 | 0.18 | 0 | 0 | 0 | 1 |
| Haptophytes-6-like | 0.18 | 0.18 | 0 | 0.005 | 0.23 | 0.47 | 0 | 0 | 0 | 1 |

**Final ratios**

| Maud Rise | Chl_c3 | Chlc_1-2 | Peri | But-fuco | Fuco | Hex-fuco | Allo | Lut | Chl_b | Chl_a |
|---|---|---|---|---|---|---|---|---|---|---|
| Prasinophytes | 0 | 0 | 0 | 0 | 0 | 0 | 0 | 0.006 | 0.533 | 1 |
| Chlorophytes | 0 | 0 | 0 | 0 | 0 | 0 | 0 | 0.239 | 0.157 | 1 |
| Cryptophytes | 0 | 0.163 | 0 | 0 | 0 | 0 | 0.191 | 0 | 0 | 1 |
| Diatoms-1 | 0 | 0.101 | 0 | 0 | 0.624 | 0 | 0 | 0 | 0 | 1 |
| Diatoms-2 | 0.187 | 0.561 | 0 | 0 | 0.974 | 0 | 0 | 0 | 0 | 1 |
| Dinoflagellates-1 | 0 | 0.221 | 0.714 | 0 | 0 | 0 | 0 | 0 | 0 | 1 |
| Dinoflagellates-2 | 0.100 | 0.284 | 0 | 0.227 | 0.588 | 0.304 | 0 | 0 | 0 | 1 |
| Haptophytes-6-like | 0.495 | 0.809 | 0 | 0.003 | 0.557 | 0.404 | 0 | 0 | 0 | 1 |

| Astrid Ridge surface | Chl_c3 | Chlc_1-2 | Peri | But-fuco | Fuco | Hex-fuco | Allo | Lut | Chl_b | Chl_a |
|---|---|---|---|---|---|---|---|---|---|---|
| Prasinophytes | 0 | 0 | 0 | 0 | 0 | 0 | 0 | 0.006 | 0.507 | 1 |
| Chlorophytes | 0 | 0 | 0 | 0 | 0 | 0 | 0 | 0.260 | 0.153 | 1 |
| Cryptophytes | 0 | 0.179 | 0 | 0 | 0 | 0 | 0.211 | 0 | 0 | 1 |
| Diatoms-1 | 0 | 0.112 | 0 | 0 | 1.232 | 0 | 0 | 0 | 0 | 1 |
| Diatoms-2 | 0.015 | 0.324 | 0 | 0 | 0.429 | 0 | 0 | 0 | 0 | 1 |
| Dinoflagellates-1 | 0 | 0.219 | 0.802 | 0 | 0 | 0 | 0 | 0 | 0 | 1 |
| Dinoflagellates-2 | 0.031 | 0.209 | 0 | 0.142 | 0.256 | 0.576 | 0 | 0 | 0 | 1 |
| Haptophytes-6-like | 0.943 | 0.392 | 0 | 0.012 | 0.502 | 0.795 | 0 | 0 | 0 | 1 |

| Astrid Ridge deep | Chl_c3 | Chlc_1-2 | Peri | But-fuco | Fuco | Hex-fuco | Allo | Lut | Chl_b | Chl_a |
|---|---|---|---|---|---|---|---|---|---|---|
| Prasinophytes | 0 | 0 | 0 | 0 | 0 | 0 | 0 | 0.007 | 0.475 | 1 |
| Chlorophytes | 0 | 0 | 0 | 0 | 0 | 0 | 0 | 0.220 | 0.136 | 1 |
| Cryptophytes | 0 | 0.156 | 0 | 0 | 0 | 0 | 0.226 | 0 | 0 | 1 |
| Diatoms-1 | 0 | 0.088 | 0 | 0 | 1.014 | 0 | 0 | 0 | 0 | 1 |
| Diatoms-2 | 0.016 | 0.276 | 0 | 0 | 0.463 | 0 | 0 | 0 | 0 | 1 |
| Dinoflagellates-1 | 0 | 0.233 | 0.765 | 0 | 0 | 0 | 0 | 0 | 0 | 1 |
| Dinoflagellates-2 | 0.035 | 0.219 | 0 | 0.263 | 0.170 | 0.723 | 0 | 0 | 0 | 1 |
| Haptophytes-6-like | 0.728 | 0.240 | 0 | 0.007 | 0.379 | 0.336 | 0 | 0 | 0 | 1 |





| Other stations | Chl_c3 | Chlc_1-2 | Peri | But-fuco | Fuco | Hex-fuco | Allo | Lut | Chl_b | Chl_a |
|---|---|---|---|---|---|---|---|---|---|---|
| Prasinophytes | 0 | 0 | 0 | 0 | 0 | 0 | 0 | 0.007 | 0.400 | 1 |
| Chlorophytes | 0 | 0 | 0 | 0 | 0 | 0 | 0 | 0.306 | 0.096 | 1 |
| Cryptophytes | 0 | 0.190 | 0 | 0 | 0 | 0 | 0.236 | 0 | 0 | 1 |
| Diatoms-1 | 0 | 0.088 | 0 | 0 | 1.030 | 0 | 0 | 0 | 0 | 1 |
| Diatoms-2 | 0.017 | 0.378 | 0 | 0 | 0.608 | 0 | 0 | 0 | 0 | 1 |
| Dinoflagellates-1 | 0 | 0.238 | 0.695 | 0 | 0 | 0 | 0 | 0 | 0 | 1 |
| Dinoflagellates-2 | 0.301 | 0.414 | 0 | 0.358 | 0.403 | 0.573 | 0 | 0 | 0 | 1 |
| Haptophytes-6-like | 0.418 | 0.280 | 0 | 0.010 | 0.189 | 1.063 | 0 | 0 | 0 | 1 |


*Peri: peridinin; Fuco: fucoxanthin; Allo: alloxanthin; Lut: lutein.*
**4. Discussion**
**4.1 Community patterns at the regional scale**
The early autumn phyto- and protozooplankton community composition in Kong Håkon VII Hav was dominated
by diatoms and other algae from the Chl *c* -lineage, which is typical for the open Southern Ocean (e.g. Davidson
et al., 2010; Kang and Fryxell, 1993; van Leeuwe et al., 2015; Nöthig et al., 2009; Peeken, 1997; Smetacek et al.,
2004; Wright et al., 2010). Some differences in the relative abundance of the major taxa were observed between
the sampling areas, which will be discussed below. When it comes to biodiversity, phytoplankton species
richness was similar between the areas investigated. The Maud Rise bloom had lower diversity indices, which
can be attributed to the dominance of *C. dicheata* during the bloom (Vallina et al., 2014) and hence is likely not
reflecting persistent lower diversity at Maud Rise compared to the other areas – both species richness and
evenness in abundances between species are components of biodiversity. The diversity index and species
richness sampling area averages in our study were clearly higher than cluster averages in a community
composition study conducted at 30° – 80° E in austral summer (Davidson et al., 2010), and the diversity indices
were relatively high for the low biomass level compared to a global data compilation (Irigoien et al., 2004).
Surprisingly, including the haptophytes pigment type 6 (“type species” coccolithophore *Gephyrocapsa huxleyi*,
formerly known as *Emiliania huxleyi*; Bendif et al., 2019) gave better results (lower error) in the preliminary
CHEMTAX analysis than including the pigment type 8 (e.g. *Phaeocystis*), and when including both pigment
types, type 6 was clearly more prominent. However, coccolithophores are not abundant this far south in the
Southern Ocean (Balch et al., 2016; Saavedra-Pellitero et al., 2014; Trull et al., 2018), which is confirmed in our
microscopy analysis. A few stations in the flow cytometry data may have had low abundances of
coccolithophores (not shown; based on high side-scattering and red fluorescence) but neither of these data
indicated a strong presence of this group throughout the study. Although blooms of *P. antarctica* are a prominent
feature in the marginal ice zones of the Ross Sea (Arrigo et al., 1999) and the Weddell Gyre (Vernet et al. 2019),
*P. antarctica* or other prymnesiophytes were not abundant in our microscopy samples. This is consistent with the
observation that blooms of *P. antarctica* are generally rare in the land-remote ACC (Smetacek et al. 2004) and
further supported by the low contribution of *P. antarctica* to bloom biomass in iron fertilization experiments



conducted in the iron-limited Southern Ocean (Boyd et al. 2008). Even the LOHAFEX iron fertilization
experiment conducted in low silicate waters with a significant seed population of small initial *P. antarctica*
colonies did not result in a bloom of this species, presumably because of strong top down control by copepod
grazers (Schulz et al., 2018). Furthermore, blooms of *P. antarctica* seem to coincide with the sea ice retreat and
ice edge (Davidson et al., 2010; Kang and Fryxell, 1993; Vernet et al., 2019). Our sampling effort was conducted
later in the season (i.e., early autumn, at the onset of sea ice formation) and could therefore partly explain why
the species was observed at low abundances. A subsequent cruise along the 6° E transect area earlier in the
season (in December 2020–January 2021) observed higher abundances of *P. antarctica* (S. Moreau et al.,
unpublished data).
Given the low contribution of both coccolithophores and *P. antarctica*, we have called the pigment group we
included in the final CHEMTAX analysis as "Haptophytes-6 -like" to acknowledge that the exact identity of this
group is unclear and can contain other types of algae that have similar pigment ratios than the haptophyte 6
group. The microscopy analysis indicated that the majority of the flagellates were different types of unidentified
flagellates in the size group 3 to 7 µm (note however that this group may and likely did also contain
heterotrophic flagellates). It should also be noted that due to the similarity in pigments and pigment ratios, this
pigment group will also contain silicoflagellates and chrysophytes. The former have a characteristic appearance
and should have been reliably identified in the microscopy samples, thus their share in the pigment group should
be correspondingly low as in the microscopy abundances. Unidentified chrysophytes on the contrary could have
formed a considerable share of this pigment group. Chrysophytes were regularly observed in our microscopy
samples, albeit not in high abundance. Unfortunately, pigment to Chl *a* ratio data are lacking for this group in the
Southern Ocean. Cryptophytes, that were relatively abundant among flagellates in the microscopy samples, also
contain similar pigments to haptophytes, but due to the low concentrations of their marker pigment alloxanthin
they do not show up strongly in the CHEMTAX results. The discrepancies might be partly explained with the
relatively small volume filtered (typically 1 L) for HPLC samples during this study, potentially leading to
underestimation of pigments that are present in trace amounts. Thus, we recommend a higher filtration volume
for further studies. All in all, our pigment composition was very similar (though with lower maximum
concentrations) than in the study by Gibberd et al. (2013) that was conducted mainly at the prime meridian and
the Weddell Sea in January – February one decade earlier.
Finally, picophytoplankton was not abundant in the area compared to nanophytoplankton – maximum
picophytoplankton abundance was 15 % of maximum nanophytoplankton abundance, and only at certain
stations, a distinct picophytoplankton occurrence was observed in the FCM biplots. The absence of coccoid
cyanobacteria in the area contributes to low picophytoplankton abundance. Likewise, Rembauville et al. (2017)
observed low picophytoplankton contribution (<20 % contribution to phytoplankton carbon) in the Indian sector
in the Southern Ocean based on bio-optical observations from biogeochemical Argo floats, however the study
area was further north than ours (around 50° S).
**4.2 Vertical patterns**
Some of the data types and analyses indicated that the phytoplankton communities differed along the depth
gradient, in addition to the spatial variability discussed in the next sections. Besides differences in biomass or


abundances (e.g., at Astrid Ridge the highest abundances were located in the upper 40 m), the species richness
was significantly lower below 40 m. In the cluster analysis (Fig. 5), a separation along sampling depth gradient
was visible in the figure (most notably separating the 25 m and 75 m depth categories), though further statistical
tests didn't indicate large differences between communities at different depths. These patterns seem to suggest
that the phytoplankton communities above and below the MLD (the average for all the stations was 36 ±13 m,
Kauko et al., 2021) differed to some degree. As species richness correlated positively with biomass (Fig. 6d),
which is a typical global pattern up to certain biomass level (Vallina et al., 2014), it is not surprising that species
richness was lower at depth when surface biomass is typically higher. However, if other abundance patterns
contributed to the depth separation was not easy to detect, as the species counts for the most abundant taxa in
depth categories (Fig. A6 and A7) did not seem to differ to a great degree from the whole station or area
averages (Fig. 2). A study from the Indian sector of the Southern Ocean concluded that phytoplankton
communities at the deep Chl *a* maximum were not fundamentally different from surface mixed layer
communities (Gomi et al., 2010), similarly to a study conducted between 30 and 80° E (Davidson et al., 2010).
Moreover, the distinct sub-surface communities dominated by large diatoms found in the Southern Ocean are
suggested to be linked to upstream surface blooms (Baldry et al., 2020).
At Maud Rise, vertical patterns were less clear as it seemed that the surface bloom was sinking based, e.g., on
relatively high Chl *a* concentrations at depth and below the MLD (Kauko et al., 2021) and dampened
diadinoxanthin vertical patterns compared to the other areas (Fig. 8f). This indicates that cells deeper in the water
column had recently been exposed to upper water column light conditions. Furthermore, the diatom community
at 100 m depth (at station 110) was dominated by *C. dichaeta*, whereas at 70 m at the same station the diatom
community was more diverse (Fig. A8). There could be a somewhat separate community below the MLD (60 m
at this station; Kauko et al., 2020), having access to more iron than the surface community and therefore thriving
there (Baldry et al., 2020), which the sinking surface bloom could be "passing by" and then again dominating at
100 m depth.

### 4.3 *Chaetoceros dichaeta* blooms associated with natural iron fertilization

The different analyses – microscopic identification and pigments (especially fucoxanthin patterns and
CHEMTAX results) – all show that a diatom bloom occurred at Maud Rise and station 53. The maximum
diatom abundance was somewhat higher compared to a study in the north-western Weddell Sea in the same
season (March): $1.9 \times 10^6$ cells $L^{-1}$ in our study compared to $1.2 \times 10^6$ cells $L^{-1}$ in Kang and Fryxell (1993). Both
blooms observed in this study were dominated by *C. dichaeta,* which is an important and widespread species in
the pelagic communities across the Southern Ocean (reviewed in Assmy et al., 2008). Maximum *C. dichaeta*
abundance of $1.6 \times 10^6$ cells $L^{-1}$ was again higher than in the above mentioned study ($0.4 \times 10^6$ cells $L^{-1}$; Kang
and Fryxell, 1993). This species seemed to belong to the diatoms pigment type 2, which was the most abundant
of all groups and had maximum values at station 53 and Maud Rise. Likewise, in the study by Wright et al.
(2010) east of our study area (30° – 80° E) the diatom type 2 was more widespread than the type 1 (though not
linked to *C. dichaeta* dominance; Davidson et al., 2010), contrary to large parts of the prime meridian area and
the Weddell Sea (Gibberd et al., 2013).
The observed bloom type belongs to the typical ecosystem of the open ocean iron-depleted areas of the Southern
Ocean, where a few large, heavily silicified species are the main bloom-forming species (Lasbleiz et al., 2016;
Smetacek et al., 2004). Grazing from copepods and protozoans exerts a strong selective pressure in these areas,
and large diatom species with strong silicate armour and spines can more easily escape predation (Hansen et al.,
1994; Irigoien et al., 2005; Löder et al., 2011; Pančić and Kiørboe, 2018; Smetacek et al., 2004). Indeed, small
copepods (180–1000 µm) and protists were the main zooplankton groups in the area and more abundant at Maud
Rise than in the other sampling areas (corresponding data for station 53 are lacking; Kauko et al., 2021).
Furthermore, amongst the diatoms characteristic of the iron-limited ACC, *C. dichaeta* seems to be quite
responsive to elevated iron levels as it dominated blooms induced by iron fertilization experiments EIFEX and
SOFEX south conducted in high silicate waters of the Southern Ocean during late austral summer (Assmy et al.,
2013; Coale et al., 2004).
The observed phytoplankton community type is in contrast to iron-replete near-coastal areas where blooms are
dominated by smaller and often spore-forming neritic diatoms e.g. from the genus *Thalassiosira* and the
subgenus *Hyalochaete* within the genus *Chaetoceros* that can realize fast growth rates (Armand et al., 2008;
Lasbleiz et al., 2016; Smetacek et al., 2004). Species belonging to these genera were observed in our samples,
but only in low abundances. Although there are regional differences in bloom magnitude and, likely, iron input
in our study area (Kauko et al., 2021; Moreau et al., in prep.), the iron input does not seem to be sufficient and
persistent enough to sustain the coastal diatom communities characteristic of the iron-replete areas of the
Southern Ocean. In this context also the inoculum is important, that is, coastal diatom species are likely to have
low seeding abundance in oceanic waters at the start of the growth season, especially the spore forming taxa that
tend to overwinter as resting spores on the seafloor. Indeed, the spore forming diatom *C. debilis* responded with
exponential growth to iron fertilization in the EisenEx experiment in the polar frontal zone of the ACC but
remained a minor component of the iron-induced diatom bloom because it started with a very low seed
population (Assmy et al. 2007). Changes in the spatial extent of the iron-replete productive system and the iron-
deplete HNLC system are reflected in diatom frustules preserved in Southern Ocean sediments covering the last
glacial and interglacial time periods. During the more iron-rich glacial periods resting spores of the above
mentioned *Chaetoceros* species dominated while the typical HNLC diatom *F. kerguelensis* dominated sediments
representative of the interglacial period with less iron input to the Southern Ocean (Abelmann et al., 2006).
The blooms in our area were likely fuelled by upwelling-induced natural iron fertilization: at Maud Rise, the sea
mount topography is suggested to lead to upwelling of nutrients (von Berg et al., 2020; Jena and Pillai, 2020;
Kauko et al., 2021; de Steur et al., 2007), whereas in the area represented by station 53 wind patterns create
suitable upwelling conditions and supply the area with additional, deep iron (Moreau et al., in prep.). Carbon
export to the deep sea is typically low in the HNLC areas of the Southern Ocean while silica export is high due
to the heavily silicified frustules of the dominant HNLC diatom taxa (Assmy et al., 2013; Smetacek et al., 2004).
On the other hand, significant carbon export from open-ocean fertilized blooms has been observed (Smetacek et
al., 2012) and attributed to mass mortality and aggregation of chain-forming oceanic *Chaetoceros* species,
particularly *C. dichaeta* (Assmy et al., 2013). In our study, the vertical Chl *a* profiles show that at Maud Rise the
biomass, as Chl *a* concentration above 0.01 mg m$^{-3}$, seemed to be sinking to approximately 300 m depth at the
time of sampling (Kauko et al., 2021). Krill (which would be an important grazer of these large and spiny



colonies; Smetacek et al., 2004) was not observed in notable abundances at Maud Rise during the cruise (Kauko
et al., 2021), which may indicate lower grazing pressure on the bloom and support vertical export as the main
loss term. Indeed, fluxes of labile organic matter to the seafloor are elevated at Maud Rise compared to the
surrounding waters (Sachs et al., 2009). On the contrary, at station 53 grazing presumably by krill played an
important role for the bloom fate (Moreau et al., in prep.).
In addition to the diatom dominance, larger nanophytoplankton (Nanophytoplankton 2 in the FCM results) were
a notable component of the community at Maud Rise and station 53 (unlike in the other sampling areas). None of
the flagellate groups identified with microscopy correlated well with these results so the identity is unknown.
Lastly, ciliates also showed patterns that were seemingly connected to the blooms and/or the nanophytoplankton
patterns, namely the larger share of tintinnid ciliates at Maud Rise and station 53.

### 539    4.4 Dominance of pennate diatoms at Astrid Ridge

Astrid Ridge and station 54 differed from the other sampling areas most notably by the more prominent role of
pennate diatoms (56 to 72 % of total diatom abundance). Phytoplankton abundance was in general much lower at
Astrid Ridge and station 54 than at Maud Rise, but diatoms were still more abundant than flagellates. The
phytoplankton community at Astrid Ridge was likely in a post bloom situation (Kauko et al., 2021). Also in this
area many of the dominant species fit into the concept of large, heavily silicified diatoms of the iron-deplete
areas (see discussion in the previous section; Smetacek et al., 2004), and *C. dichaeta* was also an important
species here. In terms of average abundance in all Astrid Ridge samples, the six most abundant taxa were the
pennate diatoms *Pseudo-nitzschia* spp., *Fragilariopsis nana*, *F. kerguelensis* and *Thalassiothrix antarctica* and
the centric diatoms *Thalassiosira* spp. and *C. dichaeta*.
Pennate diatoms are typically dominant in sea ice (Hop et al., 2020; van Leeuwe et al., 2018; Leu et al., 2015;
Poulin et al., 2011). This was also true for our study, where two ice cores sampled along the 6° E transect
showed strong dominance of pennate diatoms (≤95 % of diatom abundance; Fig. A13). Furthermore, out of the
20 dominant diatom species or genera in the ice cores and at Astrid Ridge (average of the samples down to 100
m), 12 were shared between these two habitats (Table B2; see the table also for ice core method descriptions). It
is however difficult to say whether the sea ice communities influenced the phytoplankton community
composition, as observed in spring e.g. at the West Antarctic Peninsula (van Leeuwe et al., 2020), or if the sea
ice reflected the water column community, but with some species succession towards ice specialists (Kauko et
al., 2018), as species exchange between the habitats occurs both during sea ice melt and sea ice formation
(Hardge et al., 2017). If the former was the case here, the later sea ice retreat at Astrid Ridge compared to many
of the other sampling areas (Kauko et al., 2021) could introduce algae from the sea ice at a later stage in the
growing season and possibly partly explain the dominance of pennate diatoms in this area. Due to the long sea
ice period, sea ice algae could also have a prominent sediment seed bank in the area, which could introduce cells
higher up in the water column through local current processes such as the strong tidal currents in this area
(Kauko et al., 2021). This topic thus requires further study and is interesting also in the light of any possible
costal to offshore gradients.
Astrid Ridge was most thoroughly sampled from all the sampling areas with a large number of CTD stations and
samples, with some variation seen within this area. In particular a few stations on the western part of Astrid



Ridge showed distinct features, including the highest picophytoplankton abundances and peridinin
concentrations of the entire sampling area. Future studies concentrating on the detailed current or food web
patterns in this area could indicate which processes contributed to these observations. However, when the
different parts of Astrid Ridge (southern, northern, western and eastern parts of the cross transect) were marked
in the cluster analysis using microscopy counts (figures not shown), no clear patterns emerged, and the areas
were mixed.
**4.5 A flagellate-dominated post-bloom community**
Both FCM, pigment and microscopy data indicated that flagellates and the smaller nanophytoplankton were an
important component of the phytoplankton community at the 6° E transect. According to the microscopy data,
flagellates numerically dominated over diatoms, and the observed marker pigments pointed towards a diverse
flagellate community. Except cryptophytes, flagellates remained to a large degree unidentified in the microscopy
samples, but pigment data showed that algae from the Chl *c*- lineage were most abundant. These could have been
haptophytes and possibly in addition chrysophytes (see Discussion section 4.1). Chl *b* containing algae were
present in low concentrations.
The 6° E transect area, similarly to Astrid Ridge, typically experiences summer blooms, and the low biomass and
abundances during this cruise likely point to a post-bloom situation (Kauko et al., 2021). Indeed, the importance
of flagellates and pico- and nanophytoplankton is thought to be the typical situation e.g. in the Weddell Gyre
(Vernet et al., 2019) or in the Southern Ocean in general (Buma et al., 1990; Detmer and Bathmann, 1997;
Smetacek et al., 2004) outside the bloom periods, during which larger cells, mainly diatoms, dominate. The
abundance of nanophytoplankton in our FCM samples was very similar to the suggested "background
concentration" of $2–4 \times 10^6$ cells L$^{-1}$ for the Southern Ocean (Detmer and Bathmann, 1997). Previous studies
from Wright et al. (2010) and Davidson et al. (2010) observed somewhat further east of our study area (30° – 80°
E) that the northern areas with most advanced blooms and likely depleted iron concentrations were dominated by
nanoflagellates, and suggested that krill grazing contributed to the community composition as they are
ineffective in feeding on the smaller organisms, as also pointed out by other studies (Granéli et al., 1993;
Kopczynska, 1992). Kauko et al. (2021) hypothesized that blooms in our study area were at least partly
terminated by krill grazing, as macronutrient concentrations in the upper water column were still sufficient to
support phytoplankton production during the cruise (i.e. after the peak bloom), and short-term incubations
indicated minimal iron limitation in the southern cruise area (Singh et al., in prep.).
Although station 53 was close to the 6° E transect, it showed a different relative community composition, which
could be a result of the different bloom phase. The station 53 area typically has a late bloom according to a
phenology analysis using satellite Chl *a* remote sensing data (Kauko et al., 2021) and was also during the cruise
in an earlier bloom phase than the surrounding areas. These two areas were also separated by an oceanographic
front (Moreau et al., in prep.). It can be speculated that the 6° E transect area had earlier experienced a *C.*
*dichaeta* dominated bloom similar to Maud Rise and station 53 just north of this transect, as *C. dichaeta* had
fairly high relative abundance (21 %) among diatoms along the 6° E transect.
There was possibly a south to north gradient visible in the diatom community along the 6° E transect (Fig. A14).
The relative abundance of *C. dichaeta* increased at the northernmost station, i.e., towards station 53, whereas the





relative abundance of e.g. *F. nana* decreased. Additionally, lutein and hex-fuco showed higher pigment to Chl *a*
ratios in the southern part of the transect. At the coast, several oceanographic features and processes can affect
iron sources and the phytoplankton growth environment: the Antarctic Slope Current, glacial melt-related
processes, shallower bottom topography and the occurrence of latent heat polynyas (e.g. Arrigo and van Dijken,
2003; Dinniman et al., 2020; Dong et al., 2016). Differences between onshore and offshore communities have
been observed east of the study area (between 30 and 80° E; Davidson et al., 2010). Future studies where
sampling very close to the coast is possible will give further insights into the community composition in these
areas. Due to heavy sea ice conditions, it was not possible to reach the coast during this cruise.
**5. Conclusions**
In this study, we have explored the phytoplankton community composition in a poorly studied area east of the
prime meridian in the Southern Ocean, in the Kong Håkon VII Hav. The results indicate that the area has a
typical open-ocean community composition with large, heavily silicified diatoms forming the blooms. These
species traits are according to the literature a long-term evolutionary response to the heavy grazing pressure
exerted by the micro- and mesozooplankton in the Southern Ocean. Furthermore, seasonal succession and bloom
phase differences likely contributed to differences between the sampling areas, with post-bloom areas having a
higher relative contribution by flagellates. Grazing (especially by krill) on bloom-forming species had likely
shaped the community composition. The transient diatom blooms overlay a more stable flagellate-dominated
background community.
The blooms described here were likely fuelled by natural iron fertilization driven by topography and wind-driven
upwelling. Open ocean blooms triggered by local iron input cannot rival the more productive coastal systems of
the Southern Ocean but enhance carbon export and feed a significant krill subpopulation. These results thus
indicate that there exists a "middle ground" between the iron-replete coastal blooms and the iron-deplete status
of the HNLC areas: oceanic blooms that are formed by some of the HNLC diatoms, particularly *C. dichaeta*,
with important implications for the strength of the biological carbon pump and transfer to higher trophic levels in
these areas. Compared to the neritic diatoms of the more productive coastal areas, *C. dichaeta* is a slow growing
species, but within the diatoms characteristic of the HNLC areas it is among the faster growing ones, responding
strongly to artificial (and natural) iron fertilization and contributing to carbon export. Thus, within this group, *C.*
*dichaeta* can be characterized as a bloom-former and carbon sinker.
It is important to note that while the main groups of the phytoplankton community were revealed by the pigment
data, the resolution of pigment data is not high enough to differentiate between, for instance, different diatoms
and delineate the patterns discussed above. Therefore, microscopy data or other imaging techniques are needed
to determine microphytoplankton to species level in order to fully understand the community composition. It is
also noteworthy that the pigment approach may not capture a large part of the dinoflagellate community with a
peridinin-based pigment type, as in our study the majority of dinoflagellates belonged to the genus
*Gymnodinium*, which contains similar pigments to e.g. diatoms and haptophytes and no peridinin (Jeffrey et al.,
2011). In addition, non-pigment containing heterotrophic species call for different approaches to identify this
important group. Finally, the haptophyte-type pigment group requires other types of analyses to be properly





identified. A possible solution for future studies could be a combination with 18S rRNA-sequencing, for a better
interpretation of the various target groups.
This is the first thorough characterization of phytoplankton community composition in the area, studying the
early autumn season. Future studies will show how it relates to the different seasons such as the early bloom
phase in spring and whether seasonal succession can be seen in the community composition. In addition, the
very near coast and coastal polynyas could not be sampled during this study and could potentially differ in their
community composition, and future sampling can offer further insights into possible north-south gradients.
**6. Data availability**
The data presented in this study can be found in online repositories (Norwegian Polar Data Centre,
data.npolar.no) in Moreau et al. (2020) and Kauko et al. (2022).
**7. Author contributions**
HMK planned the study, analysed the data and wrote the first manuscript draft. SM, HMK, TRK and AS planned
and carried out the field work. HMK and AS analysed the FCM samples. PA contributed with expert knowledge.
IP processed the pigment samples data and guided on the CHEMTAX analysis. MR and JW analysed the
microscopy samples. GB arranged the FCM analysis and processed the data. All authors contributed to the
manuscript writing.
**8. Competing interest**
The authors declare that they have no conflict of interest.
**9. Acknowledgements**
The Southern Ocean Ecosystem cruise 2019 was led by the Norwegian Polar Institute, with further financial
support from the Norwegian Ministry of Foreign Affairs. The Research Council of Norway (grant number
288370) and National Research Foundation, South Africa (grant UID 118715) project in the SANOCEAN
Norway–South Africa collaboration contributed to this study.
We are thankful to the captain and crew of the RV Kronprins Haakon, Nadine Steiger and John Olav Vinge for
help with water sampling, Elzbieta Anna Petelenz for supervising the flow cytometry measurements and Sandra
Murawski and Lea Phillips for technical assistance with the HPLC measurements.



**10. Appendices**
**Appendix A. Supplementary figures.**

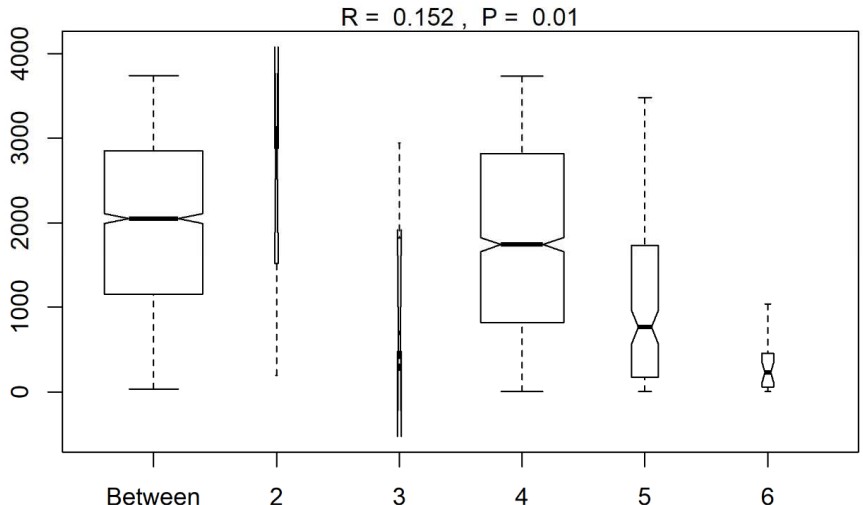


**Figure A1: A summary plot from the *anosim* analysis (testing differences between the sampling areas in species**
**abundances after the NMDS analysis). Range of dissimilarities in the different areas (2-6: station 53, station 54, Astrid**
**Ridge, the 6° E transect and Maud Rise, respectively).**





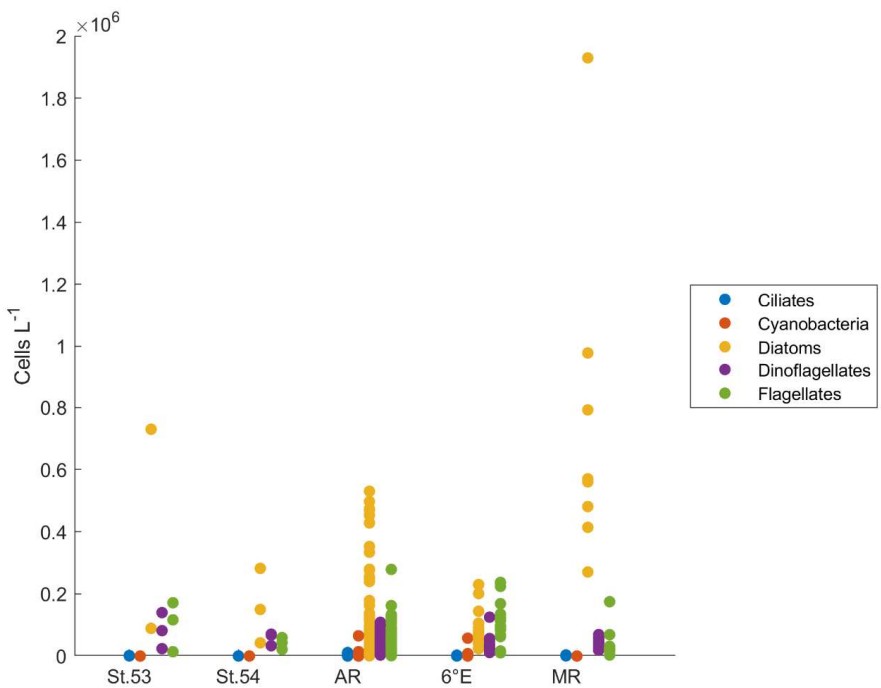


**Figure A2: Protist abundance in all samples in the different sampling areas based on microscopy. St.53=station 53,**
**St.54=station 54, AR=Astrid Ridge, 6°E= 6° E transect, MR=Maud Rise.**

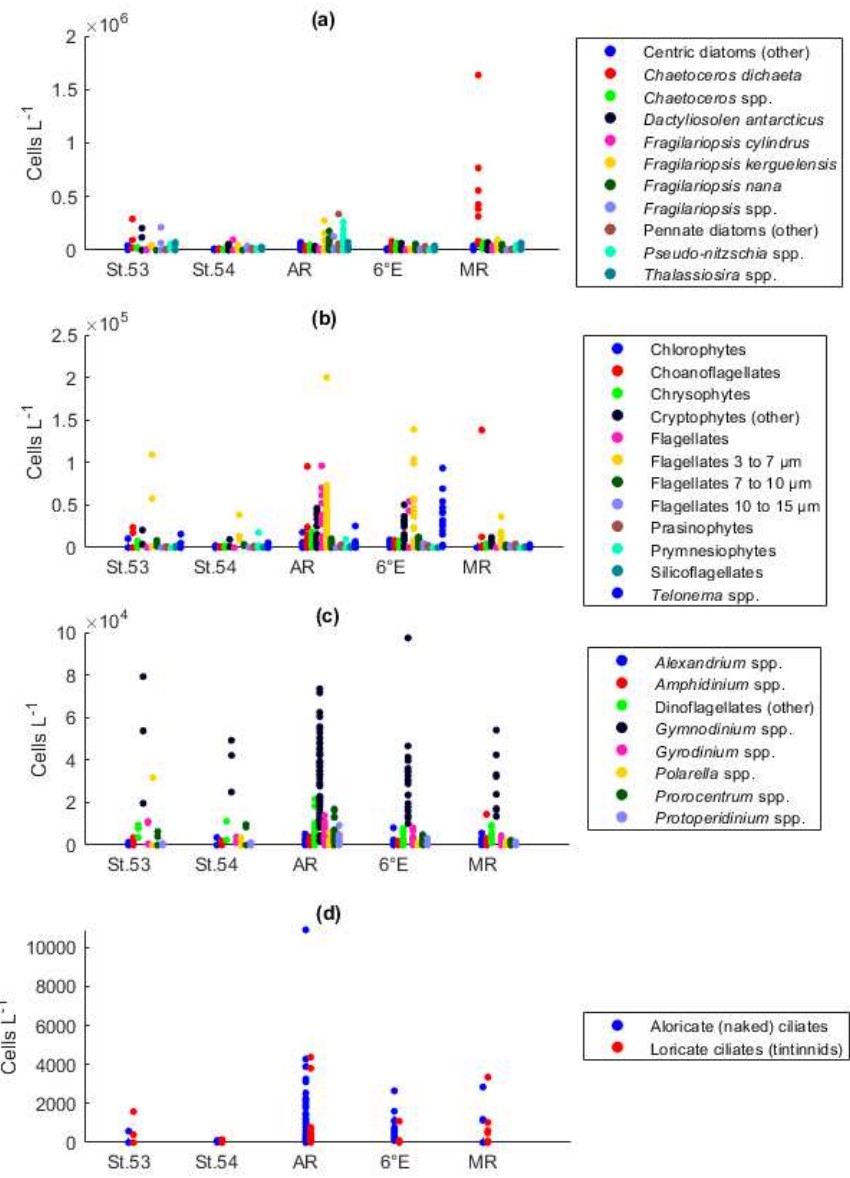


**Figure A3: Protist abundance in all samples in the different sampling areas (based on microscopy) for (a) diatoms, (b) flagellates, (c) dinoflagellates and (d) ciliates. St.53=station 53, St.54=station 54, AR=Astrid Ridge, 6°E= 6° E transect, MR=Maud Rise.**




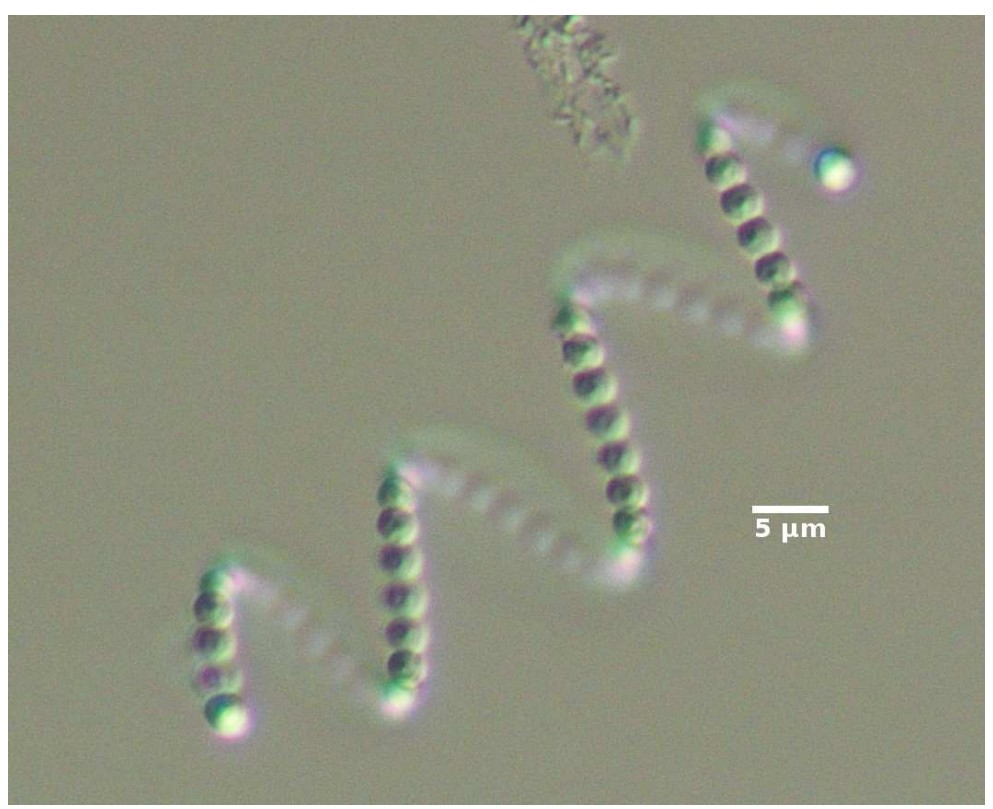


**Figure A4: Filamentous blue-green algae cf. *Anabaena* sp..**

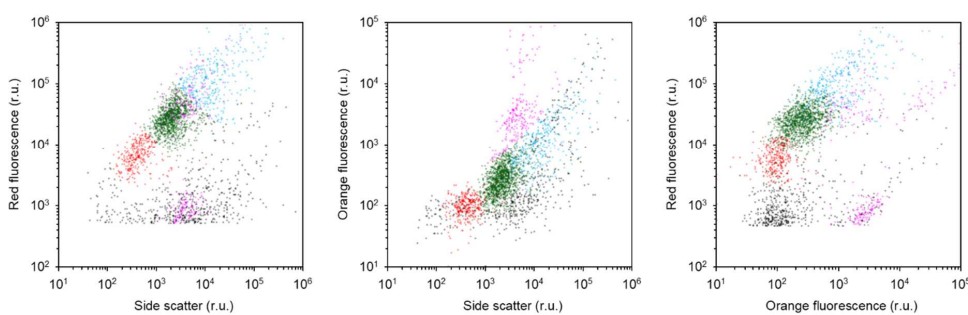


**Figure A5: Scatter plots indicating the position of the different phytoplankton populations in the cytograms. Picophytoplankton, Nanophytoplankton 1 and Nanophytoplankton 2 were discriminated based on chlorophyll red autofluorescence versus side scatter (red, green and blue dots respectively). Possible cyanobacteria and cryptophytes were in addition recognized based on their orange autofluorescence (violet dots). The example shown is from CTD station 61 at 40 m depth. Axis are in relative units (r.u.).**




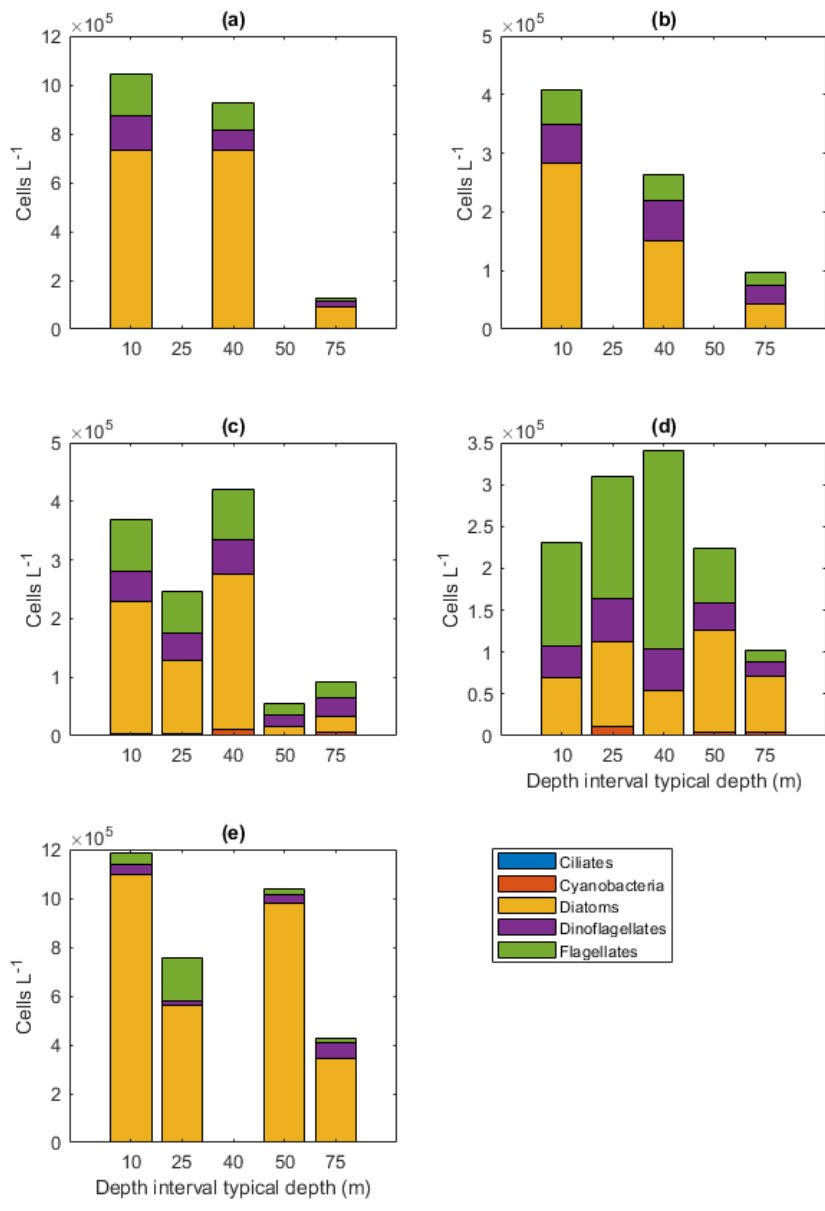


**Figure A6: Protist abundances in the different sampling areas averaged per depth interval for (a) station 53, (b) station 54, (c) Astrid Ridge, (d) 6° E transect and (e) Maud Rise. Depth intervals (with typical sampling depth in brackets): 5-10 (10); 25-35 (25), 35-45 (40), 50-60, 65-85 (75) m.**



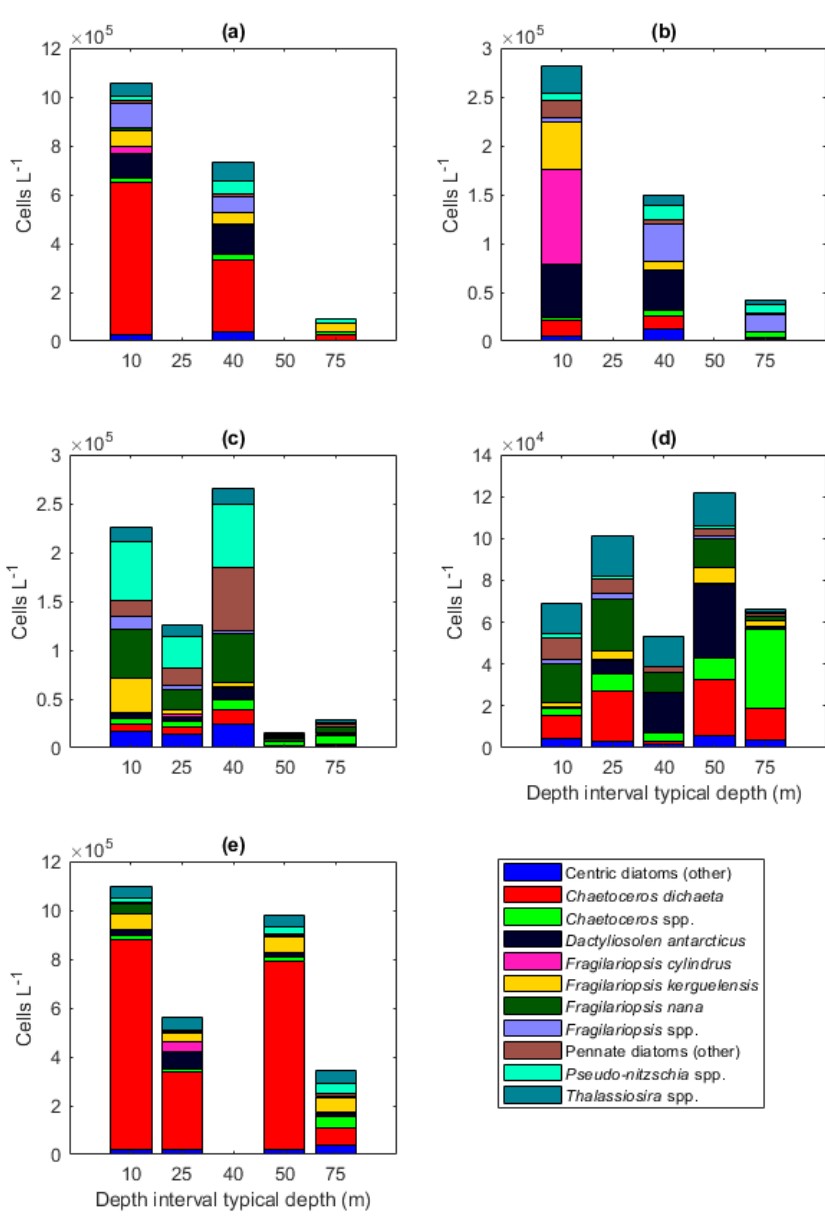


**Figure A7: Diatom abundance in the different sampling areas averaged per depth interval for (a) station 53, (b) station 54, (c) Astrid Ridge, (d) 6° E transect and (e) Maud Rise. Depth intervals (with typical sampling depth in brackets): 5-10 (10); 25-35 (25), 35-45 (40), 50-60, 65-85 (75) m.**



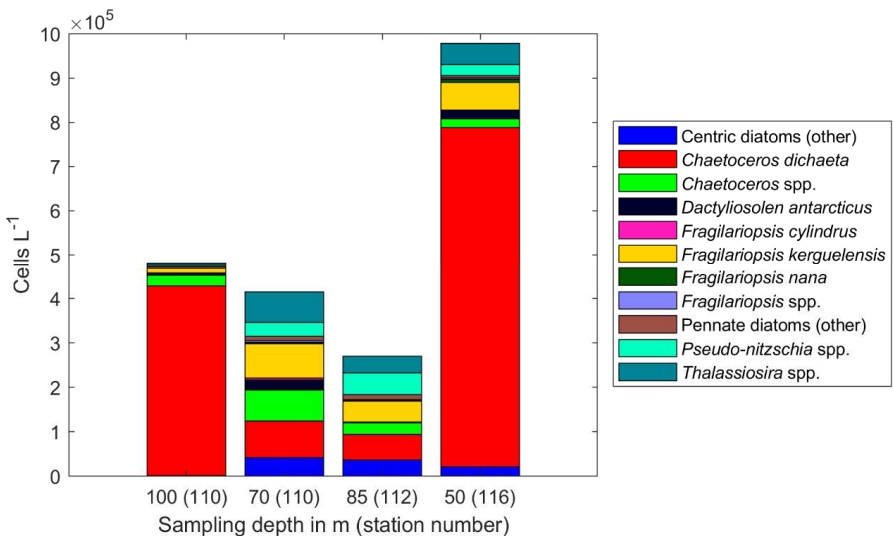


**Figure A8: Diatom abundance in available deep samples at Maud Rise. Bars are marked with the sampling depth in meters and the station number in brackets.**

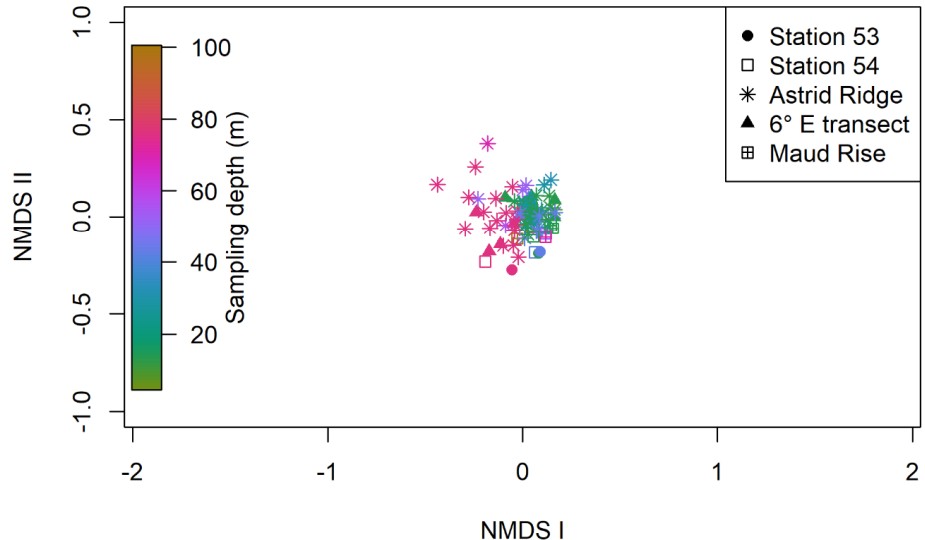


**Figure A9: NMDS clustering using presence-absence data.**



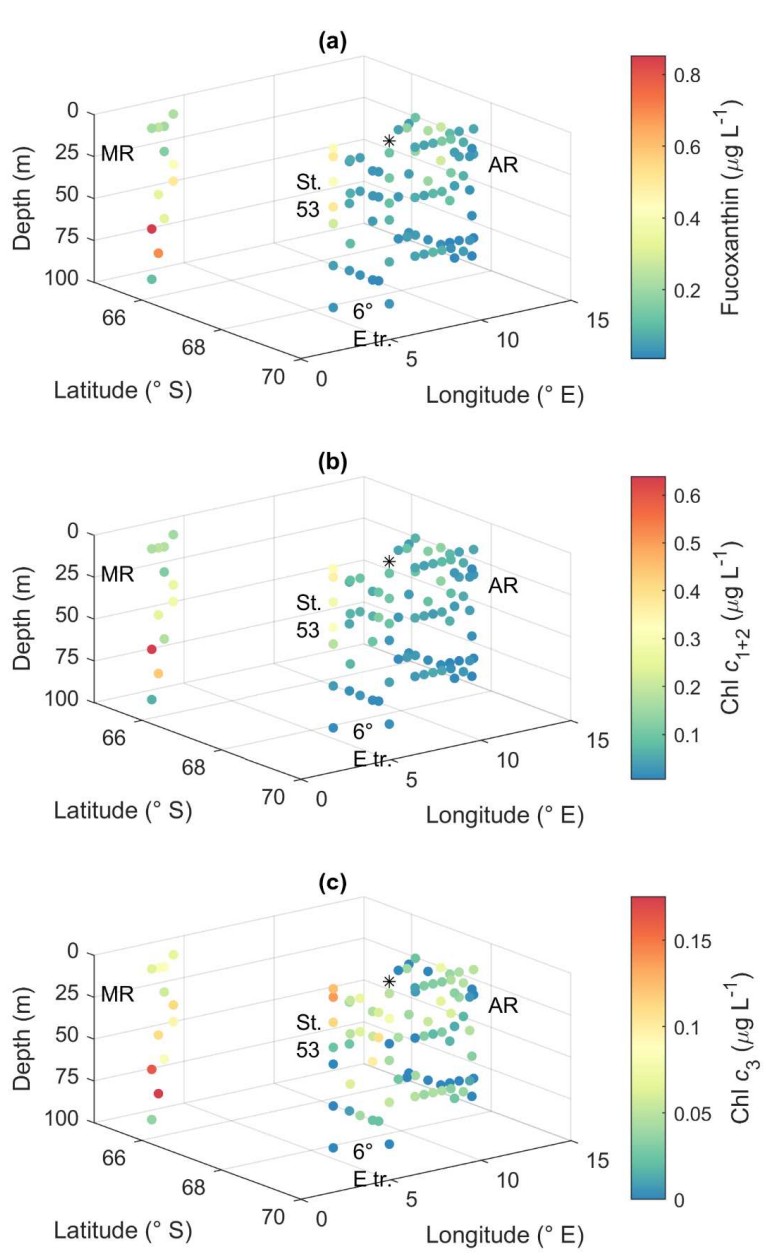



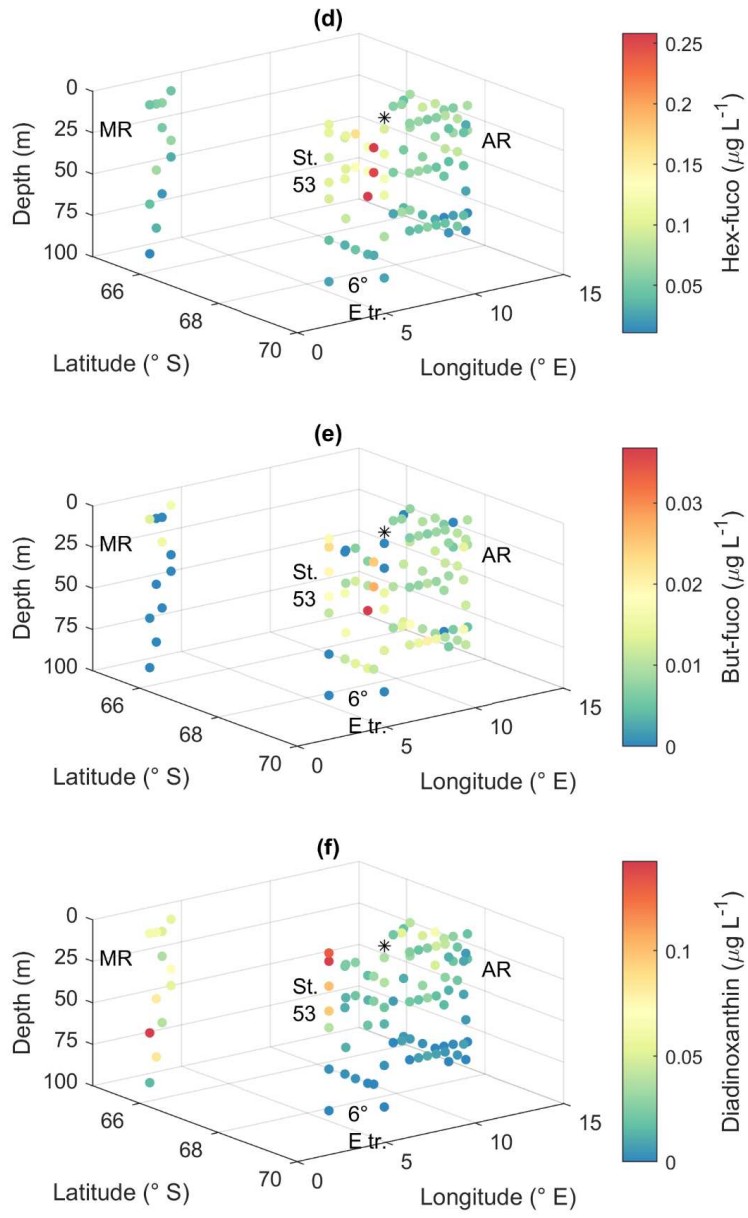


**Figure A10: Pigment concentrations of (a) fucoxanthin, (b) Chl $c_{1+2}$, (c) Chl $c_3$, (d) hex-fuco, (e) but-fuco and (f) diadinoxanthin. MR=Maud Rise, St. 53=station 53, AR=Astrid Ridge, 6° E tr.= 6° E transect. Station 54 is marked with a black asterisk.**



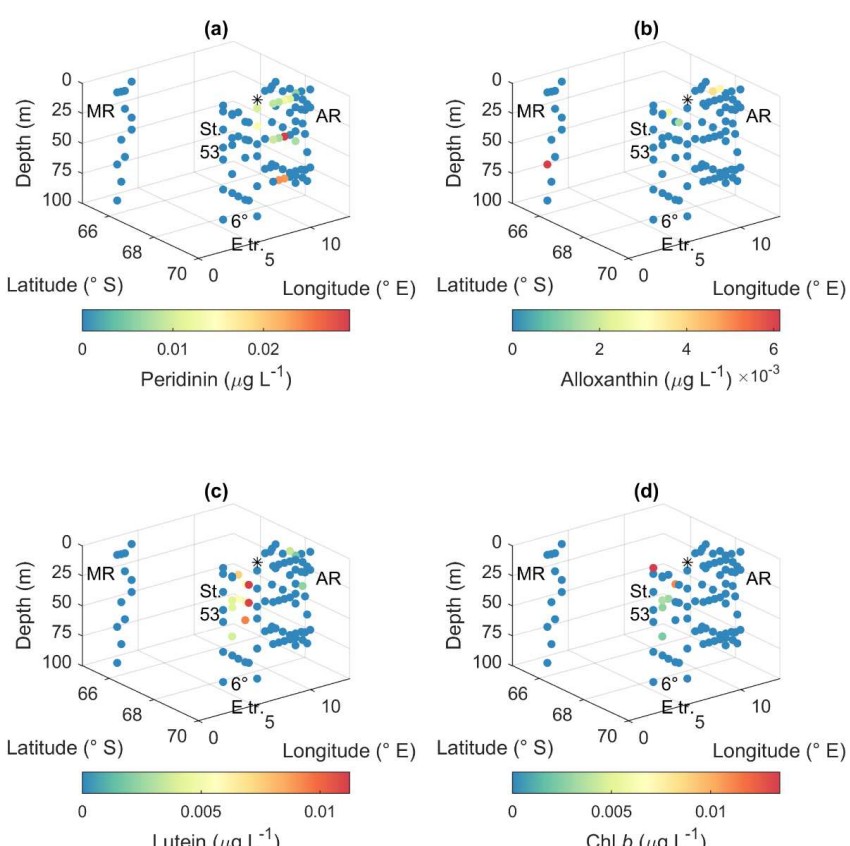


**Figure A11: Pigment concentrations of (a) peridinin, (b) alloxanthin, (c) lutein and (d) Chl *b*. MR=Maud Rise, St.**
**53=station 53, AR=Astrid Ridge, 6° E tr.= 6° E transect. Station 54 is marked with a black asterisk.**



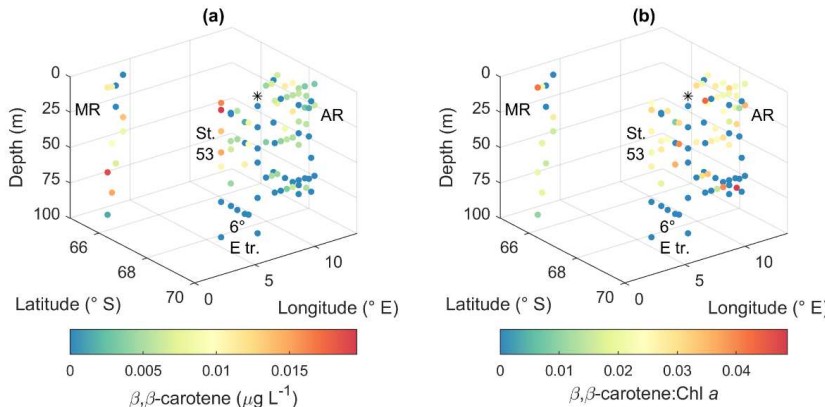


**Figure A12: (a) β ,β-carotene concentration and (b) ratio of β,β-carotene to Chl a. MR=Maud Rise, St. 53=station 53, AR=Astrid Ridge, 6° E tr.= 6° E transect. Station 54 is marked with a black asterisk.**

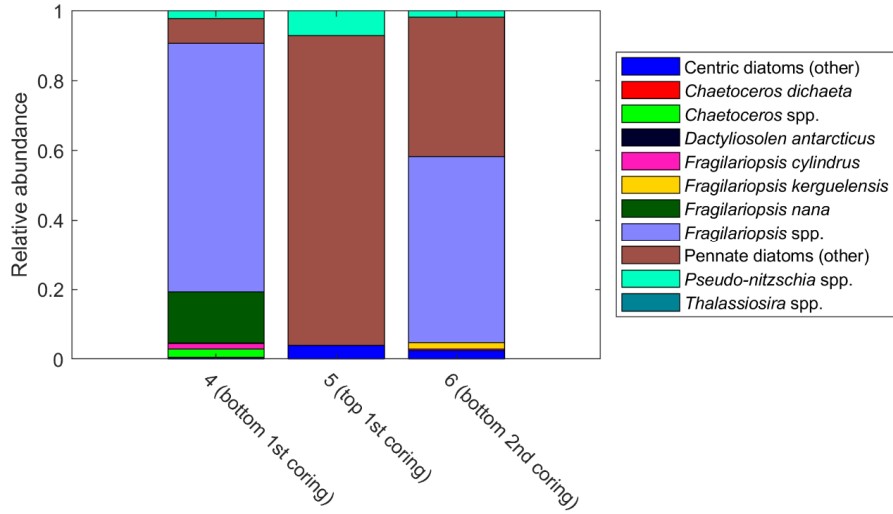


**Figure A13: Relative diatom abundance in ice core samples. The colours pink to cyan comprise pennate diatoms. The bars are marked with sample numbers and ice core section explanations. See Table B2 for method descriptions.**





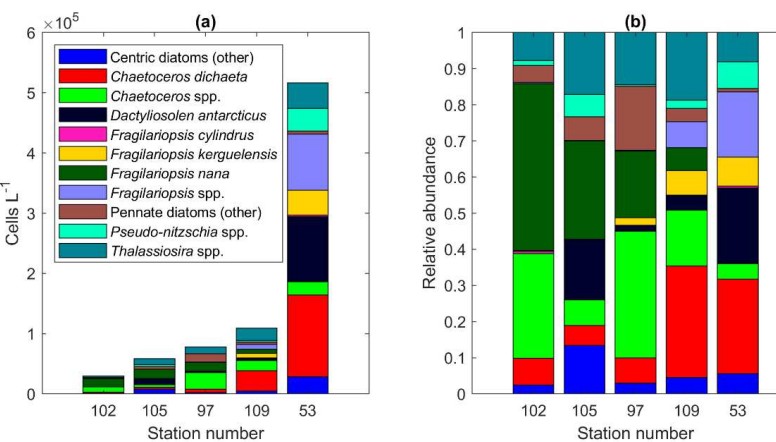


**Figure A14: (a) Diatom abundance and (b) relative abundance in the south-north transect at 6° E including the**
**station 53 just north of the transect (average abundances per station).**





**Appendix B. Supplementary tables.**

**Table B1.** All taxa identified in the CTD station samples down to 100 m (in total 87 samples). For median abundance 2, only the samples where the species/taxon was observed were taken into account (i.e., zero abundances do not contribute to the median value).

| Class/group | Species/taxon | Number of samples observed in | Median abundance 1 (cells L⁻¹) | Median abundance 2 (cells L⁻¹) | Station 53 | Station 54 | Astrid Ridge | 6° E transect | Maud Rise |
|---|---|---|---|---|---|---|---|---|---|
| Bacillariophyceae | *Actinocyclus* sp. | 1 | 0 | 2411 | | | | | |
| Bacillariophyceae | *Actinocyclus actinochilus* | 19 | 0 | 95 | x | | x | x | x |
| Bacillariophyceae | *Actinocyclus curvatulus* | 3 | 0 | 1404 | | | x | | x |
| Bacillariophyceae | *Asteromphalus* spp. | 34 | 0 | 293 | | | x | x | x |
| Bacillariophyceae | *Asteromphalus hyalinus* | 51 | 297 | 2119 | x | x | x | x | x |
| Bacillariophyceae | *Asteromphalus parvulus* | 50 | 302 | 1113 | x | x | x | x | x |
| Bacillariophyceae | *Auricula compacta* | 7 | 0 | 378 | | | x | | x |
| Bacillariophyceae | *Banquisia belgicae* | 36 | 0 | 373 | x | | x | x | x |
| Bacillariophyceae | *Chaetoceros* spp. | 55 | 1261 | 4558 | x | x | x | x | x |
| Bacillariophyceae | *Chaetoceros affinis* | 1 | 0 | 7798 | | | x | | |
| Bacillariophyceae | *Chaetoceros atlanticus* | 33 | 0 | 866 | x | x | x | x | x |
| Bacillariophyceae | *Chaetoceros atlanticus f. bulbosus* | 42 | 0 | 510 | x | | x | x | x |
| Bacillariophyceae | *Chaetoceros bulbosus* | 32 | 0 | 213 | x | x | x | x | x |
| Bacillariophyceae | *Chaetoceros castracanei* | 50 | 151 | 368 | x | | x | x | x |
| Bacillariophyceae | *Chaetoceros concavicornis* | 1 | 0 | 2133 | | | x | | |
| Bacillariophyceae | *Chaetoceros convolutus* | 1 | 0 | 3562 | x | | | | |
| Bacillariophyceae | *Chaetoceros cryophilus* | 3 | 0 | 830 | | | x | | x |
| Bacillariophyceae | *Chaetoceros curvatus* | 41 | 0 | 257 | x | x | x | x | x |
| Bacillariophyceae | *Chaetoceros decipiens* | 1 | 0 | 3059 | x | | | | |






| | | | | | | | | | | |
|---|---|---|---|---|---|---|---|---|---|---|
| Bacillariophyceae | *Chaetoceros densus* | 1 | 0 | 1029 | × | | | × | | × |
| Bacillariophyceae | *Chaetoceros dichaeta* | 75 | 4594 | 6398 | × | × | × | × | × | × |
| Bacillariophyceae | *Chaetoceros flexuosus* | 1 | 0 | 872 | × | × | | | | |
| Bacillariophyceae | *Chaetoceros neglectus* | 4 | 0 | 7600 | × | | × | × | | |
| Bacillariophyceae | *Chaetoceros simplex* | 20 | 0 | 2291 | | × | × | × | × | × |
| Bacillariophyceae | *Chaetoceros socialis* | 24 | 0 | 1078 | × | × | × | × | × | × |
| Bacillariophyceae | *Corethron* spp. | 17 | 0 | 134 | | | × | × | × | × |
| Bacillariophyceae | *Corethron inerme* | 4 | 0 | 795 | × | | × | × | | |
| Bacillariophyceae | *Corethron pennatum* | 63 | 415 | 817 | × | × | × | × | × | × |
| Bacillariophyceae | Coscinodiscophycidae | 10 | 0 | 647 | × | × | × | × | × | × |
| Bacillariophyceae | *Coscinodiscus* sp. | 2 | 0 | 4509 | | × | × | × | | |
| Bacillariophyceae | *Cylindrotheca closterium* | 84 | 1387 | 1395 | × | × | × | × | × | × |
| Bacillariophyceae | *Dactyliosolen antarcticus* | 46 | 172 | 8756 | × | × | × | × | × | × |
| Bacillariophyceae | *Dactyliosolen fragilissimus* | 1 | 0 | 8312 | × | × | | | | |
| Bacillariophyceae | *Dactyliosolen tenuijunctus* | 51 | 172 | 670 | × | × | × | × | × | × |
| Bacillariophyceae | *Entomoneis* spp. | 6 | 0 | 119 | | | × | × | × | |
| Bacillariophyceae | *Entomoneis paludosa* | 35 | 0 | 402 | | | × | × | × | × |
| Bacillariophyceae | *Eucampia antarctica* | 22 | 0 | 384 | × | | × | × | × | |
| Bacillariophyceae | *Fragilariopsis* spp. | 70 | 792 | 1153 | × | × | × | × | × | × |
| Bacillariophyceae | *Fragilariopsis curta* | 1 | 0 | 22493 | | × | × | | | |
| Bacillariophyceae | *Fragilariopsis cylindrus* | 38 | 0 | 1309 | × | × | × | × | × | × |
| Bacillariophyceae | *Fragilariopsis kerguelensis* | 63 | 1771 | 6323 | × | × | × | × | × | × |
| Bacillariophyceae | *Fragilariopsis nana* | 71 | 10683 | 17244 | | | × | × | × | × |
| Bacillariophyceae | *Fragilariopsis rhombica* | 32 | 0 | 1720 | × | × | × | × | × | × |
| Bacillariophyceae | *Fragillaria* spp. | 2 | 0 | 1600 | | | × | × | × | |
| Bacillariophyceae | *Guinardia* spp. | 2 | 0 | 10059 | × | × | × | × | | × |
| Bacillariophyceae | *Guinardia cylindrus* | 44 | 76 | 368 | × | × | × | × | × | × |
| Bacillariophyceae | *Guinardia flaccida* | 1 | 0 | 584 | | | × | | | |
| Bacillariophyceae | *Haslea* spp. | 72 | 792 | 1118 | × | × | × | × | × | × |





| Class | Species | | | | | | | | | |
|---|---|---|---|---|---|---|---|---|---|---|
| Bacillariophyceae | *Haslea trompii* | 1 | 0 | 1664 | | | x | | | x |
| Bacillariophyceae | *Haslea vitrea* | 2 | 0 | 354 | | | | | | x |
| Bacillariophyceae | *Leptocylindrus mediterraneus* | 33 | 0 | 195 | x | | x | x | x | x |
| Bacillariophyceae | *Membraneis challengeri* | 25 | 0 | 396 | x | x | x | x | x | x |
| Bacillariophyceae | *Navicula* spp. | 60 | 179 | 399 | x | x | x | x | x | x |
| Bacillariophyceae | *Navicula criophila* | 1 | 0 | 1583 | x | | | | | |
| Bacillariophyceae | *Navicula directa var. directa* | 1 | 0 | 86 | | | x | | | |
| Bacillariophyceae | *Navicula transitans* | 1 | 0 | 109 | | | x | | | |
| Bacillariophyceae | *Nitzschia longissima* | 41 | 0 | 333 | | | x | x | x | x |
| Bacillariophyceae | *Odontella* sp. | 1 | 0 | 778 | | | x | | | |
| Bacillariophyceae | *Odontella weissflogii* | 1 | 0 | 176 | | | x | | | |
| Bacillariophyceae | Pennales | 59 | 302 | 757 | x | x | x | x | x | x |
| Bacillariophyceae | Phaeoceros | 4 | 0 | 516 | x | | x | | x | |
| Bacillariophyceae | *Plagiotropus gaussii* | 1 | 0 | 938 | | | x | | | |
| Bacillariophyceae | *Proboscia* spp. | 12 | 0 | 221 | x | x | x | | | x |
| Bacillariophyceae | *Proboscia alata* | 61 | 169 | 378 | x | x | x | x | x | x |
| Bacillariophyceae | *Proboscia inermis* | 29 | 0 | 172 | x | x | x | x | x | x |
| Bacillariophyceae | *Proboscia truncata* | 6 | 0 | 315 | | | x | | | |
| Bacillariophyceae | *Pseudo-nitzschia* spp. | 78 | 1474 | 1887 | x | x | x | x | x | x |
| Bacillariophyceae | *Pseudo-nitzschia heimii* | 28 | 0 | 3392 | x | x | x | x | x | |
| Bacillariophyceae | *Pseudo-nitzschia lineola* | 13 | 0 | 1245 | x | x | x | x | x | |
| Bacillariophyceae | *Pseudo-nitzschia turgidula* | 1 | 0 | 1105 | | | | | x | |
| Bacillariophyceae | *Pseudo-nitzschia turgiduloides* | 1 | 0 | 2010 | | | x | | | |
| Bacillariophyceae | *Rhizosolenia* spp. | 25 | 0 | 165 | x | x | x | x | x | x |
| Bacillariophyceae | *Rhizosolenia delicatula* | 1 | 0 | 792 | x | | x | | | |
| Bacillariophyceae | *Rhizosolenia hebetata* | 3 | 0 | 396 | x | | x | x | | |
| Bacillariophyceae | *Rhizosolenia hebetata f. semispina* | 19 | 0 | 137 | x | x | x | x | x | x |
| Bacillariophyceae | *Rhizosolenia imbricata* | 25 | 0 | 218 | x | x | x | x | x | x |
| Bacillariophyceae | *Rhizosolenia simplex* | 2 | 0 | 534 | | | x | | | |



| Class | Species | | | | | | | | |
|---|---|---|---|---|---|---|---|---|---|
| Bacillariophyceae | Synedropsis spp. | 36 | 0 | 1505 | | | | × | × |
| Bacillariophyceae | Thalassiosira spp. | 80 | 7296 | 9321 | × | × | × | × | × |
| Bacillariophyceae | Thalassiosira frenguelli | 1 | 0 | 28817 | × | × | | | |
| Bacillariophyceae | Thalassiosira gracilis | 11 | 0 | 6560 | × | × | × | | × |
| Bacillariophyceae | Thalassiosira nordenskioeldii | 1 | 0 | 804 | | | × | | |
| Bacillariophyceae | Thalassiosira oliveriana | 1 | 0 | 396 | | | × | | |
| Bacillariophyceae | Thalassiosira perpusilla | 1 | 0 | 19418 | | | × | | |
| Bacillariophyceae | Thalassiothrix spp. | 4 | 0 | 458 | | | | | × |
| Bacillariophyceae | Thalassiothrix antarctica | 14 | 0 | 491 | × | × | × | × | × |
| Bacillariophyceae | Trachyneis aspera | 1 | 0 | 1180 | × | | × | | |
| Bacillariophyceae | Trichotoxon reinboldii | 6 | 0 | 384 | × | × | | | × |
| Bacillariophyceae | Tropidoneis sp. | 1 | 0 | 7619 | | | × | | |
| Chlorophyceae | Chlorophyceae | 1 | 0 | 10479 | × | | | | |
| Choanoflagellatea | Bicosta spinifera | 15 | 0 | 1210 | | | × | × | |
| Choanoflagellatea | Choanoflagellatea | 41 | 0 | 2310 | × | × | × | × | × |
| Choanoflagellatea | Monosiga sp. | 1 | 0 | 3251 | | | × | | |
| Choanoflagellatea | Monosiga marina | 13 | 0 | 2376 | × | × | × | × | × |
| Choanoflagellatea | Parvicorbicula socialis | 5 | 0 | 23577 | × | × | × | | × |
| Chrysophyceae | Chrysophyceae | 63 | 2140 | 3670 | × | × | × | × | × |
| Ciliophora | Amphorides laackmanni | 8 | 0 | 175 | | | × | × | |
| Ciliophora | Balanion spp. | 27 | 0 | 165 | | | × | × | |
| Ciliophora | Ciliophora | 53 | 105 | 348 | × | × | × | × | × |
| Ciliophora | Didinium spp. | 2 | 0 | 198 | | | × | | |
| Ciliophora | Lohmanniella oviformis | 20 | 0 | 188 | | | × | × | × |
| Ciliophora | Mesodinium pulex | 2 | 0 | 190 | × | | × | | |
| Ciliophora | Mesodinium rubrum | 4 | 0 | 179 | × | × | × | | |
| Ciliophora | Oligotrichida | 1 | 0 | 174 | | | | × | |
| Ciliophora | Pelagostrombidium spp. | 10 | 0 | 131 | × | × | × | | × |
| Ciliophora | Salpingella costata | 39 | 0 | 165 | × | × | × | × | × |





| Group | Species | | | | | | | | | |
|---|---|---|---|---|---|---|---|---|---|---|
| Ciliophora | Strombidiidae | 1 | 0 | 101 | | | x | | | |
| Ciliophora | Strombidium spp. | 10 | 0 | 121 | | | x | x | | |
| Ciliophora | Strombidium conicum | 25 | 0 | 174 | | | x | x | | |
| Ciliophora | Tintinnidae | 8 | 0 | 268 | | | x | | | x |
| Ciliophora | Tintinnopsis sp. | 1 | 0 | 109 | | | | x | | |
| Ciliophora | Uronema marinum | 1 | 0 | 1046 | | | | x | | |
| Cryptophyceae | Cryptophyceae | 44 | 1014 | 4497 | x | x | x | x | x | x |
| Cryptophyceae | Cryptophyceae 3 to 7 µm | 65 | 2279 | 3361 | x | x | x | x | x | x |
| Cryptophyceae | Cryptophyceae 7 to 10 µm | 50 | 1132 | 3565 | x | x | x | x | x | x |
| Cryptophyceae | Cryptophyceae 10 to 20 µm | 10 | 0 | 1685 | | | x | x | x | x |
| Cryptophyceae | Teleaulax spp. | 10 | 0 | 1280 | | | x | x | x | x |
| Cryptophyceae | Teleaulax amphioxeia | 1 | 0 | 10849 | | | x | | | |
| Cryptophyceae | Telonema spp. | 59 | 1205 | 3052 | x | x | x | x | x | x |
| Dictyochophyceae | Dictyocha speculum | 51 | 109 | 274 | x | x | x | x | x | x |
| Dinophyceae | Alexandrium spp. | 20 | 0 | 2154 | x | x | x | x | x | x |
| Dinophyceae | Amphidinium spp. | 15 | 0 | 411 | x | x | x | x | x | x |
| Dinophyceae | Amphidinium crassum | 3 | 0 | 1180 | | | x | x | x | |
| Dinophyceae | Amphidinium hadai | 33 | 0 | 804 | x | x | x | x | x | x |
| Dinophyceae | Amphidinium longum | 1 | 0 | 1631 | | | x | x | | |
| Dinophyceae | Amphidomataceae | 3 | 0 | 2310 | x | x | x | x | | |
| Dinophyceae | Dinophyceae | 23 | 0 | 2175 | x | x | x | x | x | x |
| Dinophyceae | Dinophyceae 10 to 20 µm | 22 | 0 | 1543 | | | x | x | x | x |
| Dinophyceae | Dinophyceae 20 to 30 µm | 11 | 0 | 1623 | x | x | x | x | x | |
| Dinophyceae | Dinophyceae 30 to 40 µm | 3 | 0 | 1180 | | | x | x | | |
| Dinophyceae | Dinophysis sp. | 1 | 0 | 2455 | x | | | | | |
| Dinophyceae | Diplopsalis lenticula | 1 | 0 | 3749 | x | | | | | |
| Dinophyceae | Gymnodiniales | 5 | 0 | 1623 | | | x | x | | |
| Dinophyceae | Gymnodiniales 10 to 20 µm | 3 | 0 | 1087 | | x | | x | x | |
| Dinophyceae | Gymnodiniales 20 to 30 µm | 5 | 0 | 1608 | x | | x | x | x | x |





| Class | Species | | | | | | | | | |
|---|---|---|---|---|---|---|---|---|---|---|
| Dinophyceae | Gymnodiniales 30 to 40 μm | 2 | 0 | 2298 | | | x | | | x |
| Dinophyceae | Gymnodinium spp. | 69 | 2738 | 3361 | x | x | x | x | x | x |
| Dinophyceae | Gymnodinium galeatum | 57 | 1305 | 2936 | x | x | x | x | x | x |
| Dinophyceae | Gymnodinium gracilentum | 58 | 1167 | 2438 | x | x | x | x | x | x |
| Dinophyceae | Gymnodinium wulffii | 1 | 0 | 1066 | | | x | | | |
| Dinophyceae | Gymnodinium spp. below 10 μm | 78 | 4436 | 4839 | x | x | x | x | x | x |
| Dinophyceae | Gymnodinium spp. 10 to 20 μm | 86 | 15176 | 15309 | x | x | x | x | x | x |
| Dinophyceae | Gymnodinium spp. 20 to 30 μm | 53 | 1105 | 2455 | x | x | x | x | x | x |
| Dinophyceae | Gymnodinium spp. 30 to 40 μm | 4 | 0 | 1089 | | | x | x | x | |
| Dinophyceae | Gyrodinium spp. | 24 | 0 | 1595 | x | x | x | x | x | x |
| Dinophyceae | Gyrodinium fusiforme | 1 | 0 | 1132 | x | | | x | | |
| Dinophyceae | Gyrodinium spp. 10 to 20 μm | 37 | 0 | 2360 | x | x | x | x | x | x |
| Dinophyceae | Gyrodinium spp. 20 to 30 μm | 37 | 0 | 1631 | x | x | x | x | x | x |
| Dinophyceae | Gyrodinium spp. 30 to 40 μm | 3 | 0 | 2310 | | | x | x | | |
| Dinophyceae | Gyrodinium spp. 40 to 50 μm | 2 | 0 | 2052 | | | x | x | | |
| Dinophyceae | Heterocapsa spp. | 2 | 0 | 1632 | x | | | x | | |
| Dinophyceae | Heterocapsa triquetra | 1 | 0 | 2420 | | | x | | | |
| Dinophyceae | Lessardia elongata | 11 | 0 | 1109 | x | x | x | x | x | x |
| Dinophyceae | Peridiniales | 16 | 0 | 2262 | | | x | x | x | |
| Dinophyceae | Polarella spp. | 11 | 0 | 1492 | x | x | x | x | x | x |
| Dinophyceae | Polarella glacialis | 7 | 0 | 1305 | | | x | x | x | |
| Dinophyceae | Preperidinium perlatum | 9 | 0 | 1139 | x | x | x | x | x | |
| Dinophyceae | Pronoctiluca pelagica | 5 | 0 | 1404 | | | | | x | x |
| Dinophyceae | Prorocentrum spp. | 6 | 0 | 3865 | | | x | | x | |
| Dinophyceae | Prorocentrum balticum | 1 | 0 | 6654 | | | x | x | | |
| Dinophyceae | Prorocentrum minimum | 50 | 1087 | 2279 | x | x | x | x | x | x |
| Dinophyceae | Protoperidinium spp. | 45 | 82 | 1070 | x | x | x | x | x | x |
| Dinophyceae | Protoperidinium bipes | 3 | 0 | 198 | | | x | x | x | |
| Dinophyceae | Protoperidinium smithii | 3 | 0 | 1180 | | | x | | | |






| Group | Taxon | n | | | | | | | | |
|---|---|---|---|---|---|---|---|---|---|---|
| Dinophyceae | *Protoperidinium unipes* | 1 | 0 | 1270 | | | x | | | |
| Dinophyceae | *Torodinium* sp. | 1 | 0 | 1631 | | | x | x | | |
| Eukaryote indetermined | Eukaryote indetermined | 29 | 0 | 3527 | | x | x | x | x | x |
| Eukaryote indetermined | Eukaryote indetermined 3 to 7 µm | 62 | 9753 | 17889 | | x | x | x | x | x |
| Eukaryote indetermined | Eukaryote indetermined 7 to 10 µm | 6 | 0 | 2387 | | x | x | x | | |
| Eukaryote indetermined | Eukaryote indetermined 10 to 20 µm | 2 | 0 | 1582 | | | x | x | | x |
| Eukaryote indetermined | Spore | 19 | 0 | 1180 | | | x | x | x | x |
| Flagellates | Biflagellate | 11 | 0 | 11416 | | | x | x | x | |
| Flagellates | Biflagellate 3 to 7 µm | 63 | 4265 | 6180 | x | x | x | x | x | x |
| Flagellates | Biflagellate 10 to 15 µm | 1 | 0 | 2218 | | | | | x | x |
| Flagellates | Biflagellate heterotrophic 3 to 7 µm | 1 | 0 | 8409 | | x | | | x | |
| Flagellates | Flagellate | 14 | 0 | 24026 | | | x | x | x | |
| Flagellates | Flagellate 3 to 7 µm | 73 | 14421 | 19507 | x | x | x | x | x | x |
| Flagellates | Flagellate 7 to 10 µm | 37 | 0 | 3109 | x | x | x | x | x | x |
| Flagellates | Flagellate 10 to 15 µm | 2 | 0 | 1053 | | | | x | x | x |
| Flagellates | Fourflagellate | 1 | 0 | 2335 | | | | x | x | |
| Flagellates | Fourflagellate 3 to 7 µm | 8 | 0 | 2712 | x | | x | x | x | x |
| Flagellates | Uniflagellate | 5 | 0 | 3262 | | | x | x | x | |
| Flagellates | Uniflagellate 3 to 7 µm | 24 | 0 | 3228 | | | x | x | x | x |
| Flagellates | Uniflagellate 7 to 10 µm | 3 | 0 | 1519 | | | x | x | x | |
| Flagellates | Uniflagellate 10 to 15 µm | 1 | 0 | 4869 | | | | x | x | |
| Prasinophyceae | Prasinophyceae | 1 | 0 | 1310 | x | | | | | |
| Prasinophyceae | *Pterosperma* spp. | 24 | 0 | 1552 | x | x | x | x | x | x |
| Prokaryota | Filamentous blue-green algae cf. *Anabaena* sp. | 15 | 0 | 6765 | | | x | x | x | |
| Prymnesiophyceae | *Phaeocystis antarctica* | 3 | 0 | 9628 | | x | | x | x | |
| Pyramimonadophyceae | *Pyramimonas* spp. | 35 | 0 | 2263 | | x | x | x | x | |



**Table B2.** Comparison of the 20 most abundant diatom species between sea ice samples and Astrid Ridge
samples. Green colour indicates presence in both areas.

| Ice samples (most abundant diatoms) | Average abundance (all samples; cells L⁻¹) | | Astrid Ridge (most abundant diatoms) | Average abundance (samples down to 100 m; cells L⁻¹) |
|---|---|---|---|---|
| *Fragilariopsis* spp. | 782601 | | *Pseudo-nitzschia* spp. | 30105 |
| *Fragilariopsis nana* | 152180 | | *Fragilariopsis nana* | 27081 |
| *Cylindrotheca closterium* | 53846 | | *Fragilariopsis kerguelensis* | 13004 |
| *Pseudo-nitzschia* spp. | 25263 | | *Thalassiosira* spp. | 8164 |
| *Eucampia antarctica* | 21718 | | *Thalassiothrix antarctica* | 6068 |
| *Chaetoceros* spp. | 19298 | | *Chaetoceros dichaeta* | 5954 |
| *Fragilariopsis cylindrus* | 16473 | | *Dactyliosolen tenuijunctus* | 5823 |
| *Haslea* spp. | 11706 | | *Cylindrotheca closterium* | 4436 |
| *Synedropsis* spp. | 9547 | | *Fragilariopsis* spp. | 4389 |
| Pennales | 7604 | | *Dactyliosolen antarcticus* | 3731 |
| *Navicula* spp. | 4949 | | *Chaetoceros* spp. | 3164 |
| *Chaetoceros socialis* | 4365 | | Pennales | 1656 |
| *Entomoneis paludosa* | 3201 | | *Haslea* spp. | 1646 |
| *Fragilariopsis kerguelensis* | 2855 | | *Synedropsis* spp. | 1330 |
| *Dactyliosolen tenuijunctus* | 2828 | | *Asteromphalus hyalinus* | 1269 |
| *Banquisia belgicae* | 2466 | | *Fragilariopsis cylindrus* | 1267 |
| *Chaetoceros curvatus* | 2341 | | *Corethron pennatum* | 1235 |
| *Fragilariopsis rhombica* | 2341 | | *Pseudo-nitzschia heimii* | 1199 |
| *Corethron pennatum* | 2328 | | *Pseudo-nitzschia lineola* | 1133 |
| *Odontella* spp. | 1540 | | *Thalassiosira gracilis* | 1113 |


*Two ice floes were sampled along the 6° E transect (the first one on 26.3.2019 at 68.9135° S and 6.0217° E, and*
*the second one on 27.3.2019 at 68.4392° S and 5.9135° E). Ice algal taxonomy and abundance samples were*
*taken from in total 3 ice core sections: a 10 cm bottom section and an 8.5 cm top section from the 18.5 cm thick*
*ice core at the first ice floe, and a 10 cm bottom section from the 93,5 cm thick ice core at the second ice floe. A*
*Kovacs 9 cm corer was used, and the ice samples were melted without the addition of filtered sea water in*
*darkness and room temperature, and processed as soon as the melting was complete.*



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
