# Peer review of "First phytoplankton community assessment of the Kong Håkon VII Hav, Southern Ocean during austral autumn"

_Biogeosciences, 2022_

## Author Response (AR1)

**Responses to Reviewer 1**

This manuscript describes the phytoplankton assemblage observed in a relatively under-studied region of the Southern Ocean, based on a variety of analytical tools that together highlight the contributions made via different methods, strengths and weaknesses, and interprets these data within their broader oceanographic context. The authors do an excellent job displaying their data in ways that make sense and do an equally strong job describing patterns in the results. The results section is very detailed, and at first it was a bit slow going to get to the heart of the interpretation. However, I found myself going back to the results and the figures as I was reading the discussion. The strength of this paper lies in the discussion, where the authors key in on the most interesting parts of their complex data set. I do not recommend  changes to this manuscript, though I have some suggestions for potential future work, based on their data, and a request for publication of the microscope imagery to assist other researchers, though that may be beyond the scope of this manuscript, and more appropriate for another venue.

Thank you very much for the positive feedback and for the appreciation of our work!

Section 4.1. While light microscopy may be the most time-consuming, this tool is critical as a complement to the HPLC/CHEMTAX work, and flow cytometry. The benefit of multiple tools is shown in section 4.1, where coccoliths are suggested by the CHEMTAX, though not observed under the scope, nor are they likely to be present in high numbers so far south, as noted by the authors. In this case, the authors identify the source as "Haptophyte-6-like" – an appropriate decision and one that suggests there is more to learn in this area. I think this is an important lesson for all, that the biochemical work is best combined with old-fashioned microscopy.

Thank you for the thoughts – we agree that multiple methods are needed to give a holistic picture of the communities, as mentioned shortly in the conclusions, and that "old-fashioned" microscopy is still a very valid approach.

Section 4.2. I appreciate the difficulties in interpreting depth-related differences, whether these are related to distinct living assemblages, settling assemblages sinking through the water column, either from directly above or from upstream. Consideration of all is important, and this snapshot study simply doesn't provide all the answers – which would instead require repeated temporal sampling, either via CTD casts or via sediment trap studies, or both. This kind of work has been done in the Ross Sea, as well as other areas of the Southern Ocean, and might provide a template for future work in this less well-studied region.

Thank you for the advice. We have now also pointed out in the manuscript in the section 4.2 (currently on lines 477-478) that different sampling schemes are needed to properly resolve these patterns.

Section 4.3. *Chaetoceros dichaeta* – excellent summary of the oceanographic character of the region and the role of seeding and grazing, in guiding the diatom community – two factors that are often left out of discussion – combined with iron fertilization.

Section 4.4. Astrid Ridge pennates – long history of studies in Antarctica/Southern Ocean, besides those few referenced, that describe the relationship between sea ice algae and marginal ice zone blooms, and the dominance of pennate diatoms within sea ice. Papers by

David Garrison, Kurt Buck, Ryszard Ligowski, Sarah McGrath Grossi and Neil Sullivan - for example - might be referenced here.

Thank you for these suggestions to strengthen the literature review. We have added articles from the mentioned authors to the manuscript section 4.4 which discusses the relationship between sea ice and pelagic algal communities, as well as to the section 4.1 on general phytoplankton community structure of the region.

(line 564, coastal instead of costal)

Thank you for pointing this out, the error is now corrected.

Section 4.5 Flagellate-dominated post-bloom community – I like the "complete" phytoplankton assemblage study as presented. I think this kind of approach, looking at more than just the diatoms, is going to be increasingly important as environmental change, dominated by warming, but accompanied by factors such as changes in stratification of the upper ocean, nutrient availability, and sea ice extent and duration, becomes more and more critical in driving change in the phytoplankton community at the group level.

Thank you for the comment and acknowledgement of the work – we agree on the importance of studying the different phytoplankton groups.

General comments:

I realize this may be a big "ask" for this paper, but I think images from the inverted microscopy would be very helpful for other researchers who would like to do similar research, with most phytoplankton researchers familiar with diatom identifications, but less so with the other algal groups. This may be something for a future publication - I think it would be a great contribution.

Thank you for the suggestion. Unfortunately, we consider this beyond the scope of the manuscript, also because images were not taken systematically, but we will keep it in mind for future publications. We have included an image of the cyanobacteria present in the samples in the supplementary figures in the appendix.

Any consideration of future sediment coring to address longer-term changes in this sector? It would be a great addition to our background understanding how oceanographic and climatic changes have influenced the ecosystem over time.

Very few paleo records exist from Kong Håkon VII Hav (e.g. Forsberg et al., 2003, doi: 10.1016/s0031-0182(03)00402-4), but colleagues at the Norwegian Polar Institute are indeed planning new sediment coring in future proposals and projects.

Figure 1: contour interval for map?

This information has now been added to the figure caption.

**Responses to Reviewer 2**

The Biogeosciences submission 'First phytoplankton community assessment of the Kong Håkon VII Hav, Southern Ocean during austral autumn' by Kauko et al. shows a lot of potential. The authors have collected a nice dataset about the taxonomy and distribution of the phytoplankton in three distinct regions of the Kong Håkon VII Hav – Southern Ocean. However, the data analysis is very basic and could be greatly improved with the introduction of statistical approaches that allow linking the structure of phytoplankton communities and the environment. The presentation and discussion of results is a simple description (very subjective) of the patterns found in the study region. For example, the separation of sub regions is a great idea, but it comes across very subjective and not quantitatively based at all. How are these phytoplankton-dominated regions determined? I want to see a statistical determination of subregions. In fact, why don't you use a multi-parameter analysis and use the ancillary data, nutrients, mixed layer depth, temperature, salinity, Chl a, phytoplankton assemblage and properly determine these phytoplankton niches? These also need to be clearly mapped out – I want to exactly see these subregions and the conditions that the phytoplankton exist in.

Thank you for the appreciation of our work and for the suggestions for improvement.

The line numbers in these responses refer to the cleaned version of the revised manuscript (without tracked changes).

Phytoplankton bloom phenology and environmental settings during the cruise were presented in a previous study, Kauko et al. (2021), doi: 10.3389/fmars.2021.623856. We apologize if the links to this paper were not clear enough. In the revised version of the manuscript, we now include in the section 2.1 (which is restructured to contain only general information about the cruise) a paragraph where environmental observations are shortly described (lines 91-95). The Discussion sections already had several references to Kauko et al. (2021) where these previous results are referred to. As these results were already published, we chose not to repeat them in this paper (which in addition already has a lot of figures to show the breadth of the phytoplankton community data, both in the main text and in the appendices). Therefore, this study and Kauko et al. (2021) best function when read one after the other.

Separation of the study area into subregions was made based on patterns found in the phenology study and topographic and hydrographic features (affecting the phytoplankton blooms; see Kauko et al. 2021). Subregion assignment can be exemplified by the bloom observed at station 53 that typically occurs later in the season than elsewhere in the study area. By visually combining patterns of bloom timing and magnitude, different subregions emerged in the study area (see Figure 12 in Kauko et al., 2021). We now explain the focus on the subregions (sampling areas) in the revised version of the manuscript in the Methods section in lines 87-90. The purpose of the present paper was to study whether the phytoplankton communities in these areas differ from each other and have specific characteristics (see, e.g., in the Introduction in lines 66-68).

The NMDS analysis presented in section 3.2 and Figure 5 was an attempt to separate the sampling areas statistically based on the species abundances. While some separation was seen in this analysis, all in all the sampling areas had similar communities. This thus informs us on that the whole region has a common phytoplankton community type (and we discuss and compare this type to other regions in the Southern Ocean), but we also saw differences in the relative abundances of the main groups, which we discuss in the paper. We considered these as interesting features, although the NMDS analysis did not indicate a clear separation into different clusters.

To study the environmental control on the community composition, we originally conducted a CCA analysis with the microscopy count data in full resolution and environmental variables. The following environmental variables were investigated initially: bottom depth; latitude; macronutrient concentrations; sample depth alone and in relation to the mixed layer depth; and the abundance of different grazer groups. Because of non-independent variables or non-normal distribution, some variables were excluded, and the following ones were chosen for the CCA analysis: latitude, sampling depth distance to MLD, bottom depth, silicate, copepods in the two largest size categories, krill larvae, ciliates and protists. However, only a minor fraction of variability in the abundance data could be explained by these analyses (<20 %) and they were, therefore, not included in the previously submitted version of the manuscript. The analysis was limited by the lack of dissolved iron data (even though a parallel study suggests that phytoplankton communities were not limited by iron) and matching adult krill data (acoustic observations are only available between the stations).

We now conducted a new CCA analysis with a different approach, and following the Reviewer's suggestions, by: grouping the phytoplankton abundances into coarser groups (corresponding to the geographical community features we discuss), omitting zooplankton (and variables that indicate the location such as latitude and bottom depth), and including salinity and temperature. We included this new analysis in the Results section 3.2 and discuss it in the Discussion section 4.1 of the revised version of the manuscript (lines 422-428). The orientation of the sampling areas and environmental variables was very similar with another type of analysis explored (fitting of environmental variables on an unconstrained ordination (the NMDS analysis)), however, MLD showed to be less important than the other variables in this analysis (not shown in the manuscript for brevity and to avoid repetition). In addition, both PCA with the environmental variables and CA with the coarse taxonomic groups showed a similar pattern regarding the distribution of the areas (not shown).

I would encourage the authors to revise the manuscript and submit it again. At its present state, however, I do not feel I can recommend its publication.

**Specific comments**

- line 143 'two types of diatoms…" The HPLC method used cannot separate chlorophyll c1 from c2, so this separation into two types of diatoms becomes difficult and arbitrary. Furthermore, chlorophyll c2 is a poor taxonomic biomarker as it is present in most marine phytoplanktonic groups (red lineage).

We agree that the chemotaxonomic power of Chl c2 is reduced as it is present in many algal groups, but it is not present in chlorophytes and prasinophytes which we also include in the analysis. Please also note that the main driver for separating the two diatom types is based on the occurrence of Chl c3 (e.g., Wright et al., 2009, doi: 10.1007/s00300-009-0582-9). In addition, the amount of marker pigments included should be high enough in relation to the algal groups included (Mackey et al., 1996, doi:10.3354/meps144265).

- The chemotaxonomic characterization of dinoflagellates-2 and haptophytes-6 is also complicated by the biomarkers (ratios) used – very similar. This separation is only effectively possible if some specific biomarker pigments from each group are used (e.g., gyroxanthin diester; 4-keto-Hex-fuco; Chl c2-MGDG [14/14] and Chl c2-MGDG [18/14] – see Wright & Jeffrey 2006 [https://link.springer.com/chapter/10.1007/698_2_003] or Mendes et al. 2018 [https://doi.org/10.1016/j.dsr2.2017.12.003].

Thank you for the clarification and the additional references. We have included new text in the revised version of the manuscript to acknowledge that additional pigments would be needed to unambiguously distinguish between these groups (in section 4.1. in lines 471-474). However, the included groups are as much as possible based on microscopy. We followed the suggestion by Higgins et al. 2011 to constrain pigment ratios based on microscopic examination (in: Roy et al. (2011) Phytoplankton Pigments. Characterization, Chemotaxonomy and Applications in Oceanography). Since microscopic investigations showed the dominance of *Gymnodinium*, we considered it appropriate to include the dinoflagellates-2. Similarly, haptophytes are a typical component of the phytoplankton community in these areas and may have been abundant in the unidentified flagellate category in our samples. However, we call the group "haptophytes-6-like" as it is not clear to which group these algae belong when the HPLC results are seen in connection with microscopy results (as we discussed in section 4.1). We believe that the present study is thus valuable in showing that if one is using only HPLC to study the phytoplankton community in this area (especially with the typical set of pigments), one needs to be cautionary about the haptophyte group whose identity is not yet known.

- line 435 'Cryptophytes, … also contain similar pigments to haptophytes' This is not true, right? ... and your table 1 makes that clear.

Thank you for pointing this out, we meant to refer to certain pigment, but have removed this sentence now.

- lines 437-439 'The discrepancies might be partly explained with the relatively small volume filtered (typically 1 L) for HPLC samples…' This is also a flawed argument, because for microscopy the volume used is much smaller than for HPLC. In fact, the higher volume used for pigment samples (HPLC) is an advantage of chemotaxonomic methods.

We have removed this sentence.

- line 195 'Two of the sampling locations had an active diatom bloom…' For me, characterizing a bloom situation with chlorophyll-a values below 1 mg.m$^{-3}$ is quite strange. I understand that it is a predominantly oligotrophic region, but this delimitation of what is (or is not) a bloom will have to be better defined/discussed.

We understand this concern. We have modified the mentioned sentence in the Results section to remove the statement, and included in the Discussion section 4.3 in lines 512-518 a description of why we call these patterns a bloom in the Discussion and other conclusive parts of the manuscript. A bloom phenology study (conducted with remote-sensing data) previously showed that the average Chl a concentration during the blooms and the bloom amplitude in this area are mainly in the order of 0.5-1.5 mg Chl a m$^{-3}$ (Fig. 6c and 11c in Kauko et al., 2021). We visited the study area late in the growing season (in late March), therefore it can be anticipated that the Chl a concentration earlier in the season was higher than the one we measured, as was also suggested by the remotely sensed Chl a concentration, and that the observed maximum concentrations of >0.5 mg Chl a m$^{-3}$ indicated a seasonal phytoplankton bloom. The method used in the phenology study uses an annual relative threshold for bloom definition and therefore captures biomass above the baseline levels (see Kauko et al., 2021).

- The results section will have to be restructured and, essentially, reduced. There is redundant information that does not add content to the discussion of the data presented. I speak, for example, of the graphs with the pigmentary ratios (Figs. 8 and 9) and NMDS analyses (Fig. 5). For this, authors first need to define very well the focus they want to give to the work, as it cannot simply be an unbridled compilation of hard-to-connect data.

In response to the Reviewer's suggestion, in the revised version of the manuscript, we have moved the figure 9 to the appendices; we have moved the CHEMTAX ratio table to the appendices; we have restructured the NMDS clustering into one subchapter together with the new CCA analysis; and we have shortened the text in a few places. We hope that these changes will improve the flow of the Results section as suggested by the Reviewer.

We are, however, reluctant to make further reductions to the text and figures for the following reasons, and since Reviewer #1 valued the detailed Results section. First, we have chosen to keep the NMDS analysis to study the differences/similarity between the phytoplankton communities in the different sampling areas, as was also suggested in the Reviewer's general comments to perform statistical studies to differentiate the sampling areas. The purpose of this analysis is stated in the Methods section 2.3 in line 131, which we have slightly modified. We now also mention the NMDS analysis in the Discussion section 4.1 (in lines 420-421), in addition to section 4.2 (in lines 486-488).

Second, as mentioned in the Introduction (in lines 71-76) in connection with the objectives, we used different methods to study the phytoplankton community as each of these methods gives an incomplete picture but by combining multiple methods more comprehensive information on the community can be gained. We have now improved this description. As this is the first assessment of the microbial communities in the area, all these methods bring new insights. On the other hand, in the Discussion section, we point out on several occasions (e.g., in lines 616-617) how results from the different methods support each other (e.g., when similar results are seen with different methods (e.g., diatom dominance) giving additional confidence in the results, or when microscopy and flow cytometry data support the interpretation of CHEMTAX results). Moreover, we consider that biodiversity is an important aspect in community studies.

Similarly, and finally, as this is the first phytoplankton community assessment of this area, we think it is important to include the main marker pigments (Fig. 8) in the main text to describe the patterns found in this area. Presenting the ratios to Chl $a$ instead of the absolute pigment concentrations gives a better idea of the community composition, which is the purpose of this manuscript, and shortens the Results section (as pigment concentrations are shown in the appendix). This is now also mentioned in the beginning of the Results section 3.4 (in lines 347-348). The Results section 3.4 also describes which algal groups the different pigments indicate. Also, as a Chemtax analysis is conducted, we think it is informative to show the pigment patterns prior to that. The pigment concentration and ratio to Chl $a$ results are observations, as opposed to the derived Chemtax analysis, with the above mentioned challenges. In the Discussion sections, pigment data patterns are connected with other data types, as described above.